

# A better representation of VOC chemistry in WRF-Chem and its impact on ozone over Los Angeles

Qindan Zhu[1,2,*], Rebecca H. Schwantes[1], Matthew Coggon[1], Colin Harkins[1,2], Jordan Schnell[1,2], Jian He[1,2], Havala O. T. Pye[3], Meng Li[1,2], Barry Baker[13], Zachary Moon[13,14], Ravan Ahmadov[4], Eva Y. Pfannerstill[5], Bryan Place[3], Paul Wooldridge[6], Benjamin C. Schulze[7], Caleb Arata[8], Anthony Bucholtz[9], John H. Seinfeld[10], Carsten Warneke[1], Chelsea E. Stockwell[1,2], Lu Xu[1,2,***], Kristen Zuraski[1,2], Michael A. Robinson[1,2], Andy Neuman[1], Patrick R. Veres[1,**], Jeff Peischl[1,2], Steven S. Brown[1,11], Allen H. Goldstein[5,8], Ronald C. Cohen[6,12], and Brian C. McDonald[1]

[1]NOAA Chemical Sciences Laboratory, Boulder, CO
[2]Cooperative Institute for Research in Environmental Sciences - University of Colorado Boulder
[*]Now at Department of Earth, Atmospheric and Planetary Sciences, Massachusetts Institute of Technology, Cambridge, MA, United States
[3]ORISE at Office of Research and Development, U.S. Environmental Protection Agency, Research Triangle Park, North Carolina 27711, United States
[4]NOAA Global Systems Laboratory, Boulder, CO
[5]Department of Environmental Science and Policy Management, University of California, Berkeley, Berkeley, CA 94720, United States
[6]Department of Chemistry, University of California, Berkeley, Berkeley, CA 94720, United States
[7]Department of Environmental Science and Engineering, California Institute of Technology, Pasadena, CA 91125, United States
[8]Department of Civil and Environmental Engineering, University of California, Berkeley, Berkeley, CA 94720, United States
[9]Department of Meteorology, Naval Postgraduate School, Monterey, CA 93943, United States
[10]Department of Environmental Science and Engineering, California Institute of Technology, Pasadena, CA 91125, United States
[11]Department of Chemistry, Univesity of Colorado, Boulder, Boulder, CO 80309, United States
[12]Department of Earth and Planetary Sciences, University of California, Berkeley, Berkeley, CA 94720, United States
[13]NOAA Air Resources Laboratory, College Park, MD 20740, USA
[14]Earth Resources Technology (ERT), Inc., Laurel, MD 20707, USA
[**]Now at Research Aviation Facility, National Center for Atmospheric Research
[***]Now at Department of Energy, Environmental and Chemical Engineering, Washington University in St. Louis, MI 63130, United States

**Correspondence:** Qindan Zhu (qdzhu@mit.edu) and Rebecca Schwantes (rebecca.schwantes@noaa.gov)

**Abstract.** The declining trend in vehicle emissions has underscored the growing significance of Volatile Organic Compound (VOC) emissions from Volatile Chemical Products (VCP). However, accurately representing VOC chemistry in simplified chemical mechanisms remains challenging due to its chemical complexity including speciation and reactivity. Previous studies have predominantly focused on VOCs from fossil fuel sources, leading to an underrepresentation of VOC chemistry from

5   VCP sources. We developed an integrated chemical mechanism, RACM2B-VCP, that is compatible with WRF-Chem and is aimed to enhance the representation of VOC chemistry, particularly from VCP sources, within the present urban environment. Evaluation against the Air Quality System (AQS) network data demonstrates that our model configured with RACM2B-VCP



reproduces both the magnitude and spatial variability of $O_3$ as well as $PM_{2.5}$ in Los Angeles. Furthermore, evaluation against comprehensive measurements of $O_3$ and $PM_{2.5}$ precursors from the Reevaluating the Chemistry of Air Pollutants in California (RECAP-CA) airborne campaign and the Southwest Urban $NO_x$ and VOC Experiment (SUNVEx) ground site and mobile laboratory campaign, confirm the model's accuracy in representing $NO_x$ and many VOCs and highlight remaining biases. Although there exists an underprediction in the total VOC reactivity of observed VOC species, our model with RACM2B-VCP exhibits good agreement for VOC markers emitted from different sectors, including biogenic, fossil fuel, and VCP sources. Through sensitivity analyses, we probe the contributions of VCP and fossil fuel emissions to total VOC reactivity and $O_3$. Our results reveal that 52% of the VOC reactivity and 35% of the local enhancement of MDA8 $O_3$ arise from anthropogenic VOC emissions in Los Angeles. Significantly, over 50% of this anthropogenic fraction of either VOC reactivity or $O_3$ is attributed to VCP emissions. The RACM2B-VCP mechanism created, described, and evaluated in this work is ideally suited for accurately representing ozone for the right reasons in the present urban environment where mobile, biogenic, and VCP VOCs are all important contributors to ozone formation.

## 1   Introduction

Volatile organic compounds (VOCs) are a major contributor to air pollution in urban areas, and some VOCs are harmful to human health (Shuai et al., 2018; Alford and Kumar, 2021). Through reactions with hydroxyl radicals (OH) and nitrogen oxides ($NO_x$), VOCs fuel the formation of ozone ($O_3$) and secondary organic aerosols (SOA). Short and long-term exposure to $O_3$ and SOA have been linked to numerous adverse health effects, including cardiovascular and respiratory diseases (e.g. Chen et al., 2007; Baltensperger et al., 2008). As North American cities have seen frequent exceedances of $O_3$ and particulate matter (EPA, 2016), VOCs have been targeted for reduction as a means of improving air quality in urban areas.

Mobile sources have historically been the largest emitter of VOCs in urban areas, but emissions from vehicles have declined due to control strategies. California started early and aggressively to implement regulations reducing emissions of VOCs from vehicles. Since 1970 the California Air Resources Board (CARB) mandated auto-manufacturers to meet the nation's initial motor vehicle emission standards, aiming to control VOC and $NO_x$ emissions. As a result, VOCs in Los Angeles have decreased by almost two orders of magnitude during the past five decades (Warneke et al., 2012). However, the declining rate of VOCs has slowed in recent years (Kim et al., 2022), indicating a larger contribution of VOCs emitted from non-mobile sources. VOCs such as isoprene and monoterpene are predominantly emitted from biogenic sources (Guenther et al., 1995), which is expected to increase as a response to urban greening programs and as temperatures rise in the future due to climate change (Livesley et al., 2016; Connop et al., 2016; Gu et al., 2021). Additionally, emerging VOC sources, including volatile chemical products (VCPs), have been getting more attention. VCP sources, including solvents, adhesives, cleaning agents, pesticides, and personal care products, constitute half of the fossil fuel VOC emissions in industrialized cities (McDonald et al., 2018b). While a missing OH reactivity of 8-10 $s^{-1}$ was identified at the Pasadena ground site during the CalNex campaign in 2010, approximately 50% of the missing OH reactivity is estimated to be due to emissions of unmeasured VCPs (Hansen et al., 2021).



The increasing contribution of VOCs from non-mobile sources alters the response of local ozone formation to VOC budgets. Qin et al. (2021) shows that VCP sources produce 41% of the total photochemical organic PM2.5 and 17% of the maximum daily 8 hr average ozone in the summer in Los Angeles. Similarly, Coggon et al. (2021) shows that VCPs accounted for more than 50% of the total anthropogenic VOC contribution to ozone in New York City. The effect of biogenic VOCs from

urban vegetation on ground-level ozone has been observed in cities such as Berlin (Churkina et al., 2017) and Beijing (Ma et al., 2019). It is worth emphasizing the temperature dependence of ozone can also be altered. Nussbaumer and Cohen (2020) showed that in Los Angeles, VOCs can be categorized into two groups, one that was more independent of temperature and has been effectively controlled and one that is temperature-dependent, which is now a much larger component of the remaining VOC. While $O_3$ is determined by the relative ratio of $NO_x$ and VOCs, the response of temperature effects on ozone is nonlinear.

For instance, Nussbaumer and Cohen (2020) found a lower temperature dependence of ozone in 2014-2019 than in 1997-1998 because reduced $NO_x$ results in higher ozone at low temperatures.

Representing VOC chemistry in model simulations is challenging due to the number of VOC species and corresponding reactions. It is impossible to incorporate the explicit mechanisms of all VOCs into the 3D chemical transport model due to computational expense. To simplify this, the 3D chemical transport model uses a limited number of individual species or surro-

gates, where a group of species with similar structure and OH reactivity are lumped together into a surrogate. Previous chemical mechanisms have been simplified to primarily focus on representing the VOC chemistry from mobile sources in detail, given that emissions were historically dominated by mobile sources (Middleton et al., 1990). Consequently, an updated mechanism incorporating more complex chemistry is necessary to accurately represent the varying VOC mixture and VOC chemistry in the present urban environment. Coggon et al. (2021) used the RACM-ESRL-VCP mechanism, which added updates to oxygenated

VOC chemistry relevant to VCPs to the RACM-ESRL mechanism (Stockwell et al., 1997; Geiger et al., 2003; Kim et al., 2009; Ahmadov et al., 2012). Several advancements have been made beyond the RACM-ESRL mechanism. As an example, RACM2 (Goliff et al., 2013) was created to update ketones, aromatics, and biogenically emitted compounds. RACM2_Berkeley2.0 included updates to the isoprene oxidation mechanism and monoterpene chemistry, as well as an extended mechanism for organic nitrates (Browne et al., 2014; Zare et al., 2018).

In our study, we developed an updated chemical mechanism based on the RACM2_Berkeley2.0 mechanism, named RACM2B-VCP, which aims to better represent VOCs and VOC chemistry in the present urban environment. By starting with the RACM2_Berkeley2.0 mechanism, mobile VOCs are reasonably represented with the updates in RACM2, and biogenic VOCs are represented well with the updates in RACM2_Berkeley2.0. This work builds from this past work to add improved VCP chemistry to create RACM2B-VCP, an integrated chemical mechanism ideally suited for the present urban environment where

mobile, biogenic, and VCP VOCs all need to be accurately represented in order to simulate ozone correctly for the right reasons. We evaluate the RACM2B-VCP mechanism by comparing it against a more simplified, but established mechanism RACM-ESRL-VCP, and against airborne, mobile, and ground measurements. Due to advancements in VOC measurement techniques, unique VOC tracers emitted by VCPs can now be directly measured (Gkatzelis et al., 2020). By adding these VCP VOC tracers explicitly into the RACM2B-VCP mechanism and comparing them directly to observations, we are able to bet-

ter constrain emission inventories and identify gaps in our understanding of VCP emissions and chemistry than are possible



with more condensed mechanisms. Additionally, because the RACM2B-VCP mechanism is more complex than the RACM-ESRL-VCP mechanism, more tracers for mobile and biogenic emissions and their oxidation products are available to directly compare against observations, which enables a more complete evaluation of VOC emissions and chemistry in general. We also utilize the RACM2B-VCP mechanism to investigate the temperature dependence of ozone, which will provide insights into ozone pollution scenarios in the warming future due to climate change. In this work, we describe the development of the RACM2B-VCP chemical mechanism in Sect. 2, the WRF-Chem configuration in Sect. 3 and the observations used for model evaluations in Sect. 4. We then evaluate how WRF-Chem simulation represents $O_3$, $PM_{2.5}$ (Sect. 5), $NO_x$, CO, VOCs (Sect. 6), PAN and speciated aerosols (Sect. 7). Lastly, we investigate the impact of VCP emissions and other anthropogenic VOC emissions on VOC reactivity and $O_3$ (Sect. 8).

## 2 Development of RACM2B-VCP chemical mechanism

The RACM2B-VCP mechanism is based on the RACM2_Berkeley2.0 mechanism (Browne et al., 2014; Zare et al., 2018), and includes 8 new species (Table. S1) and 25 new reactions. In addition, 43 existing reactions are updated to reflect the latest understanding of kinetics and chemistry. The following sections describe the key updates made to RACM2B-VCP.

### 2.1 Representation of VCP VOC chemistry

Oxygenated VOCs, including methanol, ethanol, isopropanol, ethylene glycol, propylene glycol, glycerol, and acetone, compose a large fraction of VCP emissions. RACM2_Berkeley2.0 mechanism already includes the degradation of ethanol, methanol, acetone, and ethylene glycol. We add isopropanol, propylene glycol, and glycerol as new species into RACM2B-VCP. The products of OH reactions and the reaction coefficients are the same as those in RACM-ESRL-VCP, which is described in Coggon et al. (2021). Each species serves as a surrogate for other oxygenated VOCs emitted from VCPs, which are largely composed of glycols and glycol ethers. A description of the lumping schemes is provided in the supplemental information of Coggon et al. (2021).

VOC tracers linked to specific VCP categories have been identified in Gkatzelis et al. (2020), therefore we implement four tracers as separate species in order to validate the VCP emissions as model inputs. These four tracers are D4-siloxane, D5-siloxane, *p*-dichlorobenzene (PDCBZ) and *p*-chlorobenzotrifluoride (PCBTF). D4-siloxane and D5-siloxane are predominantly emitted from adhesives and personal care products, respectively. Alton and Browne (2020) and Alton and Browne (2022) determined the kinetics and products of oxidation by the OH radical. The oxidation products of both D4-siloxane and D5-siloxane are C3 and higher alcohols (ROH). D5-siloxane has a higher reactivity towards OH radicals and has a reaction constant of $2.1 \times 10^{-12}$ $cm^3 molec^{-1} s^{-1}$. In contrast, the reaction constant of D4-siloxane with OH is $1.3 \times 10^{-12}$ $cm^3 molec^{-1} s^{-1}$. *P*-dichlorobenzene and *p*-chlorobenzotrifluoride are treated as tracers for insecticides and solvent-based coatings, respectively. We assume their products with OH reactions are the same as species with the closest functional structure. We assume that *p*-dichlorobenzene oxidation by OH yields the same products as benzene, and the reaction constant is $3.2 \times 10^{-13}$ $cm^3 molec^{-1} s^{-1}$ (Atkinson and Arey, 2003). The *p*-chlorobenzotrifluoride oxidation by OH yields the same products





as toluene, and the reaction constant is $2.5 \times 10^{-13} \; cm^3 molec^{-1} s^{-1}$ (Atkinson and Arey, 2003). The addition of reactions associated with the VCP VOC chemistry in the RACM2B-VCP mechanism is summarized in Table S2.

## 2.2 Implementation of TUV photolysis scheme

The tropospheric Ultraviolet and Visible radiation model version 5.3.2 (Madronich and Flocke, 1997) is used for photolysis parameterization. While this TUV scheme had previously only been coupled with MOZART mechanisms in WRF-Chem (ACOM, 2021), we add the capability to link it to both RACM-ESRL-VCP and RACM2B-VCP mechanisms. The photolysis reactions in both mechanisms are mapped with an optional scaling factor to photolysis rate constants already incorporated into the TUV lookup table. The mapping between RACM-ESRL-VCP and TUV is described in Table S3 and the mapping between RACM2B-VCP and TUV is described in Table S4 and Table S5. For instance, the hydroperoxy aldehydes (HPALD1 and HPALD2) are not included in the TUV lookup table. We assume that the hydroperoxy aldehydes photolyze with the cross sections of methacrolein (Wennberg et al., 2018) and the quantum yield estimated by Liu et al. (2017). The organic nitrates, including ethanal nitrate (ETHLN), propanone nitrate (PROPNN), methacrolein nitrate (MACRN) and methyl vinyl ketone nitrate (MVKN), are mapped to nitrate organic aerosol (NOA) in TUV based on scaling from MCM v3.3.1 as done in Schwantes et al. (2020). All hydroxy nitrates are assumed to photolyze with the same rate of $CH_2OHCH_2ONO_2$ computed in TUV.

To better represent the photolysis feedback, we incorporate boundary layer clouds and total column ozone from Global Forecast System (GFS) Model into the TUV scheme, consistent with Rapid Refresh coupled with Chemistry (RAP-Chem) (Benjamin et al., 2016).

## 2.3 Implementation of secondary organic aerosol (SOA) scheme

Both RACM-ESRL-VCP and RACM2B-VCP mechanisms use the MADE (Modal Aerosol Dynamics for Europe) aerosol scheme (Ackermann et al., 1998). The RACM2_Berkeley2.0 mechanism inherits the SORGAM parameterization for secondary organic aerosol (SOA) (Schell et al., 2001). This parameterization has been proven to substantially underestimate the SOA production as well as the abundance of organic aerosol (OA) in urban plumes (McKeen et al., 2007). We choose to replace this outdated SOA parameterization with the new SOA_VBS parameterization (Ahmadov et al., 2012), that is used in the RACM-ESRL-VCP mechanism. This SOA scheme is based on the volatility basis set and categorizes each SOA class into four volatility bins. The VOCs forming SOA are divided into two groups, anthropogenic and biogenic. We respeciate the RACM2B-VCP species to the Ahmadov et al. (2012) scheme as shown in Table S6, but keep the same general structure and SOA yields. We note that the SOA scheme in both RACM-ESRL-VCP and RACM2B-VCP likely overestimates SOA from mobile emissions due to producing SOA from both aromatic VOCs (e.g., toluene) and their oxidation products (e.g., cresol) and underestimates SOA from VCPs, which are generally not considered as SOA precursors in this scheme. Future work will update this SOA scheme to more accurately represent SOA formation from biogenic, mobile, and VCP sources.





## 2.4 Implementation of aerosol uptake coefficients

Aerosol uptake has been proven to be a significant loss pathway for gas-phase organic nitrates from lab and field studies (Day et al., 2010; Darer et al., 2011; Hu et al., 2011; Jacobs et al., 2014; Fry et al., 2013; Teng et al., 2017). It has recently been applied to several mechanisms used in 3D chemical transport models (Schwantes et al., 2020; Zare et al., 2018, 2019; Bates and Jacob, 2019; Müller et al., 2019). In the RACM2_Berkeley2.0 mechanism, the uptake of isoprene nitrates and monoterpene nitrates are described by constant reaction rates. These reaction rates are determined using an estimated fraction of tertiary

vs. non-tertiary nitrate (Zare et al., 2018) and a timescale of 3h for tertiary nitrates, where the latter is based on a laboratory chamber study (Boyd et al., 2015). Here, we update them to a parameterization using the uptake coefficient ($\gamma$), aerosol surface area, and aerosol diameter, with the same approach as applied to the MOZART-T1 mechanism in WRF-Chem (Emmons et al., 2020). To account for the difference in the solubility, we apply an uptake coefficient of 0.005 for isoprene nitrates and 0.01 for all monoterpene nitrates following Fisher et al. (2016). Even though monoterpene nitrates are likely to lead to SOA production

(Zare et al., 2019; Pye et al., 2015), we attribute the products of aerosol uptake of organic nitrates to gas-phase nitric acid and do not account for the particle phase hydrolysis afterward.

The aerosol uptake of inorganic species is also important (Jacob, 2000). RACM2_Berkeley2.0 accounts for the $N_2O_5$ uptake, forming $HNO_3$, with a constant reaction rate. We update the $N_2O_5$ uptake and also add the aerosol uptakes of $NO_3$, $NO_2$, and $HO_2$. Following Jacob (2000), we define the $N_2O_5$ uptake with $\gamma_{(N_2O_5)}$ of 0.1, the $NO_3$ uptake with $\gamma_{(NO_3)}$ of $10^{-3}$ and

155 the $NO_2$ uptake with $\gamma_{(NO_2)}$ of $10^{-4}$. We also define the product of $NO_3$ to $HNO_3$, and the products of $NO_2$ to HONO and $HNO_3$. However, we note that other studies suggest different uptake coefficients and products for $N_2O_5$ (Brown and Stutz, 2012; Chang et al., 2016; McDuffie et al., 2018), $NO_3$ (Brown and Stutz, 2012) and $NO_2$ (VandenBoer et al., 2013, 2015). The aerosol uptake of $HO_2$ is highly uncertain. Despite evidence in some studies suggesting that $HO_2$ uptake does not occur (e.g. Tan et al., 2020), Jacob (2000) recommended that the product of $HO_2$ uptake is $H_2O_2$, and it limits the efficiency as an $HO_2$

sink as $H_2O_2$ photolyzes to regenerate OH and from there $HO_2$. Mao et al. (2013) proposed a catalytic mechanism to rapidly convert $HO_2$ to $H_2O$ in aqueous aerosols with transition metal ions (Cu and Fe). The alternative $H_2O$ formation serves as a dominant $HO_2$ loss pathway in remote environments such as the Arctic (Mao et al., 2010). Previous studies have reported a wide scattering of $HO_2$ uptake coefficients $\gamma_{(HO_2)}$, ranging from 0.1 (Christian et al., 2017) to 1.0 (Emmons et al., 2015). In RACM2B-VCP, we decided to take $H_2O$ as the product of $HO_2$ uptake and an uptake coefficient of 0.1, which is consistent

with the treatment of $HO_2$ uptake in GEOS-Chem (Christian et al., 2017) and CAM-Chem (Gaubert et al., 2020). All uptake coefficients of both organic nitrates and inorganic species are summarized in Table S7.

## 2.5 Implementation of eucalyptol and updates to reactions rate

We add a new species ECLP to represent eucalyptol. The monoterpenes measured from the proton-transfer-reaction mass spectrometer (PTR-MS) instrument could be biased high due to the interference of eucalyptol (Kari et al., 2018). Further-

170 more, eucalyptol has been shown to be an important contributor to monoterpenoid emissions in the LA basin (Van Rooy et al., 2021). Monoterpenes are measured at m/z 137.13, which may include fragments of monoterpenoids and monoterpene alcohols



whose parent mass is m/z 155.14 ($C_{10}H_{18}O$, e.g. eucalyptol). We account for the reaction of eucalyptol with OH; the products are the same as HC8 surrogate species and the rate constant is $1.1 \times 10^{-11}$ $cm^3 molec^{-1}s^{-1}$ (Corchnoy and Atkinson, 1990). Compared to alpha-pinene and limonene, whose reaction rates with OH are $5.2 \times 10^{-11}$ $cm^3 molec^{-1}s^{-1}$ and $1.6 \times 10^{-10}$ $cm^3 molec^{-1}s^{-1}$ at the temperature of 300K, eucalyptol has much lower reactivity. RACM2B-VCP now has three monoterpene-monoterpenoid surrogates that span an OH reactivity of an order of magnitude, which is especially important for evaluating total monoterpenes against aircraft observations where some more reactive monoterpenes have already declined due to chemical loss.

We also update the inorganic reaction rate to the most recent reaction rates from JPL Evaluation Number 19 (Burkholder et al., 2020). The changes are summarized in Table S8. The only exception is the reaction of $O(3P) + NO_2 - > NO + O_2$, we use the reaction rates from JPL Evaluation Number 18 since JPL Evaluation Number 19 combines the reactions of $O(3P) + NO_2 - > NO + O_2$ and $O(3P) + NO_2 - > NO_3$.

## 3  Model configuration

Both chemical mechanisms, RACM-ESRL-VCP and RACM2B-VCP, are incorporated into the 3D chemical transport model, Weather Research and Forecasting with Chemistry v4.2.2 (WRF-Chem). WRF-Chem is set up with a horizontal spatial resolution of 4km × 4km in California during the summer of 2021. It is a nested domain simulation with the initial boundary conditions constrained by a preceding 12km × 12km model simulation covering the Contiguous US. We use the Regional Air Quality Modeling System (RAQMS) as the boundary condition for the 12km model simulations (http://raqms-ops.ssec.wisc.edu/). The model configuration is explained further in Li et al. (2021) and summarized here.

The meteorological fields in the 12km model run are initiated with the North American Mesoscale Forecast System (NAM) model. In the 4km model run, we utilize the meteorological fields from the High-Resolution Rapid Refresh (HRRR) model due to a higher spatial resolution (3km) than NAM (12km). The MYNN-EDMF planetary boundary layer and shallow cloud scheme (Olson et al., 2019; Angevine et al., 2020) are used. This scheme includes vertical mixing of chemical species consistent with its mixing of physical variables.

The anthropogenic emissions here are from the FIVE-VCP-NEI17NRT inventory. This inventory is described by McDonald et al. (2018a, b) and Coggon et al. (2021) and updated by He et al. (in review) with near real-time (NRT) scaling factors capturing changes in emissions in the 2019-2021 timeframe, such as those due to the COVID-19 pandemic. Mobile source emissions are from the Fuel Based Inventory for Vehicle Emissions (FIVE) and are updated using fuel sales. VCP emissions are calculated using a mass balance of US chemical product manufacturing for 2010 (McDonald et al., 2018a; Coggon et al., 2021) and updated using economic activity scaling factors. Other sectors in the inventory are from the US Environmental Protection Agency's (EPA) 2017 National Emissions Inventory (NEI) and, where applicable, updated with near real-time scaling factors calculated from energy and economic metrics relevant to specific subsectors. The emission inventories are re-speciated for both RACM-ESRL-VCP and RACM2B-VCP mechanisms. The mapping between emission inventories and species in both mechanisms is described in Table S9. Because the RACM2B-VCP mechanism has more species, the mapping is more explicit



and requires fewer scaling factors. For instance, benzene is mapped to toluene in the RACM-ESRL-VCP mechanism. We need to apply a scaling factor of 0.29 to account for the difference in OH reactivity between benzene and toluene. In contrast, benzene and toluene are treated as separate species in the RACM2B-VCP mechanism. There is no need to apply the scaling as it is in RACM-ESRL-VCP, which improves the representation of aromatic oxidation and enables a more fair evaluation against observations.

The biogenic emissions are based on the Biogenic Emission Inventory System (BEIS) v3.14. We have considered urban vegetation as additional biogenic emissions and linked them to urban land cover types and leaf area index. For the biogenic emission used in the RACM-ESRL-VCP mechanism, we add additional isoprene and monoterpene emissions following Scott and Benjamin (2003). Among the monoterpene emissions, we assume that 20% is limonene and 80% is alpha-pinene, which is the same as the previous work over the LA region (Kim et al., 2016). We update the urban biogenic emission to include

more species in the RACM2B-VCP mechanism and account for the emission of eucalyptol. Eucalyptol constitutes a significant portion of total monoterpene emissions, ranging from 2% and 72% depending on the tree types (Owen and Penuelas, 2013; Van Meeningen et al., 2017; Zuo et al., 2017; Purser et al., 2021). Van Rooy et al. (2021) provided the most recent biogenic VOC observations in the Los Angeles Basin and found that eucalyptol comprises 10% of total monoterpene emissions. Therefore, we adjust the ratio of monoterpene emissions in accordance with Van Rooy et al. (2021), with 37% limonene, 53% alpha-pinene,

and 10% eucalyptol for the RACM2B-VCP mechanism.

## 4   Observations

Figure 1 shows a variety of observations in LA during summertime 2021 to evaluate the WRF-Chem simulations with the RACM2-VCP mechanism, including airborne, mobile, and ground measurements. The instruments for the measurements used in our study are summarized in Table S10.

The airborne measurements were conducted during the RECAP-CA (Re-evaluating the Chemistry of Air Pollutants in California) field campaign. Nine flights were conducted between 11 a.m. to 6 p.m. on June 1-22, 2021, at 300-400 m above ground. The flight tracks are shown in Figure S1(a) and are segregated into four regions following Pfannerstill et al. (2023b) and Nussbaumer et al. (2023), including Downtown LA, San Bernadino Valley, Santa Ana Valley, and Coastal LA. The instruments on board include a three-channel custom-built thermal dissociation-laser induced fluorescence (TD-LIF) to measure $NO_x$ (Zhu

et al., 2023), a Vocus PTR-ToF-MS (Proton transfer reaction time of flight mass spectrometer) to measure VOCs (Pfannerstill et al., 2023b), a Picarro G2401-m cavity ringdown spectrometer (CRDS) to measure CO and $CH_4$. The primary goal of RECAP-CA was to derive the $NO_x$ and VOC fluxes and to evaluate the emission inventory (Zhu et al., 2023; Pfannerstill et al., 2023b, a; Nussbaumer et al., 2023). In our study, we utilize the observations of CO, $NO_x$ and VOCs during RECAP-CA to compare against the WRF-Chem simulations and to evaluate the model performance in representing $NO_x$ and VOC chemistry

aloft within the mixing layer.

Additionally, the mobile laboratory and ground site measurements were conducted in August and early September 2021 as part of the SUNVEx (Southwest Urban $NO_x$ and VOC Experiment, https://csl.noaa.gov/projects/sunvex/) field campaign led



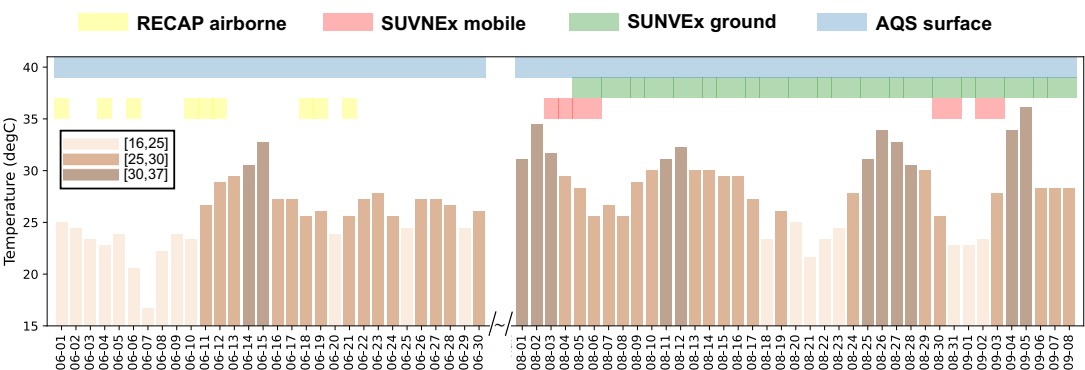

**Figure 1.** The summary of observations used for WRF-Chem evaluation spanning June 01 to September 08, 2021. It includes nine days of airborne measurements from the RECAP-CA field campaign in June 2021; eight days of mobile measurements from the SUNVEx field campaign in August and September 2021; ground site measurements from SUNVEx between August 05 and September 08, 2021; continuous surface measurements at AQS monitoring sites in June, August, and September 2021. The days are categorized into three groups based on the surface temperature at noon in Pasadena: "Low temperature" between 16 to 25 °C, "Med temperature" between 25 to 30 °C and "High temperature" between 30 to 37 °C.

by the NOAA Chemical Sciences Laboratory. Four drives were made between 7 a.m. to 7 p.m. local time at the beginning of the month between August 3-6, 2021, and four other drives were made toward the end of the month between August 30 -
September 3, 2021. The mobile drives are shown in Figure S1(b) and are segregated into the same four regions as RECAP-CA. Likewise, a ground site in Pasadena continuously monitored $NO_x$ and VOC chemistry for the whole of August and early September. Here we only include the measurements between 8 a.m. to 8 p.m. to assess the daytime chemistry. During the SUNVEx field campaign, VOCs were measured by a Vocus PTR-ToF-MS as described by Coggon et al. (2023). $NO_x$ and CO were measured by NOAA Picarro. In addition, at the ground site, PAN is measured by the NOAA Iodide Chemical Ionization
Mass Spectrometer (CIMS), and speciated aerosols are measured by the Aerosol Mass Spectrometer (AMS).

We also utilize the hourly ozone and $PM_{2.5}$ observations from the Air Quality System (AQS) monitoring network reported by the U.S. Environmental Protection Agency (EPA). One site (Ontario-Route 60 Near Road, -117.62°, 34.03°) representing the micro-scale environment and another site (Pomona, -117.75°, 34.07°) representing the middle-scale environment are filtered out as WRF-Chem at 4km cannot represent the dynamics and chemistry at the spatial scale less than 2km. After filtering, we
obtained ozone measurements at 12 sites (Table S11) and $PM_{2.5}$ measurements at 5 sites (Table S12) in the LA basin, the site map is shown in Figure S2.

The observations cover a wide range of temperatures allowing us to investigate how the model represents the chemistry in different temperature conditions. In order to normalize the temperature across the different campaigns, we use the noontime surface temperature at the Pasadena AQS site as the indicator of daily temperature and categorize the days into three tempera-
ture bins; "Low temperature" between 16 to 25 °C, "Med temperature" between 25 to 30 °C and "High temperature" between 30 to 37 °C.





## 5 Evaluation of $O_3$ and $PM_{2.5}$ against AQS network

$O_3$ and $PM_{2.5}$ are two critical criteria pollutants, and accurately representing them is crucial for the model's reliability. The RACM-ESRL-VCP mechanism and its predecessor, combined with the FIVE-VCP emission inventory, have demonstrated
effectiveness in simulating $O_3$ in the LA Basin (Kim et al., 2016; McDonald et al., 2018b; Kim et al., 2022). In particular, Kim et al. (2022) demonstrated that the WRF-Chem model using the RACM-ESRL mechanism successfully reproduced the diurnal variations and annual evolution of $O_3$ observed at 16 sites from the South Coast Air Quality Management District (AQMD) monitoring network, despite a ~14% overprediction. Here, we evaluate the performance of the WRF-Chem model configured with the RACM2B-VCP mechanism in simulating $O_3$ and $PM_{2.5}$, by comparing against not only the observations from the
AQS network but also the WRF-Chem simulation with the RACM-ESRL-VCP mechanism.

We first compare hourly $O_3$ and $PM_{2.5}$ from model simulations against the measurements in AQS sites. Figure 2 shows the comparison of the time series of hourly $O_3$ in June and August at the AQS site located at Pasadena (-118.13°, 34.13°). Both WRF-Chem simulations, configured with RACM-ESRL-VCP and RACM2B-VCP chemical mechanisms, capture the diurnal pattern and the day-to-day variation of $O_3$. The WRF-Chem with RACM-ESRL-VCP mechanism yields a normalized
mean bias (NMB) of 0.09 and a determination of coefficient ($R^2$) of 0.80. Similarly, the WRF-Chem with RACM2B-VCP mechanism yields an NMB of 0.07 and an $R^2$ of 0.81. Table S11 summarizes the comparison of hourly $O_3$ among 12 AQS sites. The WRF-Chem with RACM-ESRL-VCP mechanism reports the NMB ranging from -0.08 to 0.24 and the $R^2$ ranging from 0.51 to 0.87, and the WRF-Chem with RACM2B-VCP mechanism reports the NMB ranging from -0.12 to 0.21 and the $R^2$ ranging from 0.49 to 0.88. Figure S3 shows the comparison of the time series of hourly $PM_{2.5}$ in both months at the Ontario
AQS site (-117.62°, 34.03°), and Table S12 summarizes the comparison of hourly $PM_{2.5}$ among 5 sites. Both WRF-Chem simulations underpredict $PM_{2.5}$, the NMB is from -0.23 to -0.06 for the RACM-ESRL-VCP mechanism and is from -0.23 to -0.07 for the RACM2B-VCP mechanism. However, the model fails to capture the hourly variation of $PM_{2.5}$ observed at AQS sites with the $R^2$ ranging from 0.03 to 0.24 for both RACM-ESRL-VCP and RACM2B-VCP mechanisms.

We then compare $O_3$ and $PM_{2.5}$ at the time scale that is consistent with the National Ambient Air Quality Standards
(NAAQS), namely MDA8 (maximum 8-hour average) $O_3$ and daily (24 hours) $PM_{2.5}$ concentrations. The comparison of the spatial distribution of MDA8 $O_3$ and daily $PM_{2.5}$ is shown in Figure S4. We further categorize the sites west of longitude -117.8 ° as West/Central LA and the remaining sites as East basin. The comparison is shown in Figure 3(a) and (c) for $O_3$ and $PM_{2.5}$, respectively. Generally, the East basin exhibited significantly higher pollution levels compared to the West/Central LA, characterized by an average difference of 32 ppb in MDA8 $O_3$ and 2.7 $\mu$g/m$^3$ in daily $PM_{2.5}$ based on the AQS network
observations. Both WRF-Chem simulations successfully reproduce the positive gradient of pollution levels between west and east LA despite an overprediction in MDA8 $O_3$ and an underprediction in daily $PM_{2.5}$. WRF-Chem with the RACM2B-VCP mechanism shows slightly better results than the RACM-ESRL-VCP mechanism with respect to NMB. The NMB in MDA8 $O_3$ is 10.2% and 9.3% and instead of, the NMB in daily $PM_{2.5}$ is -13% and -11.9% for WRF-Chem with RACM-ESRL-VCP and RACM2B-VCP mechanisms, respectively.



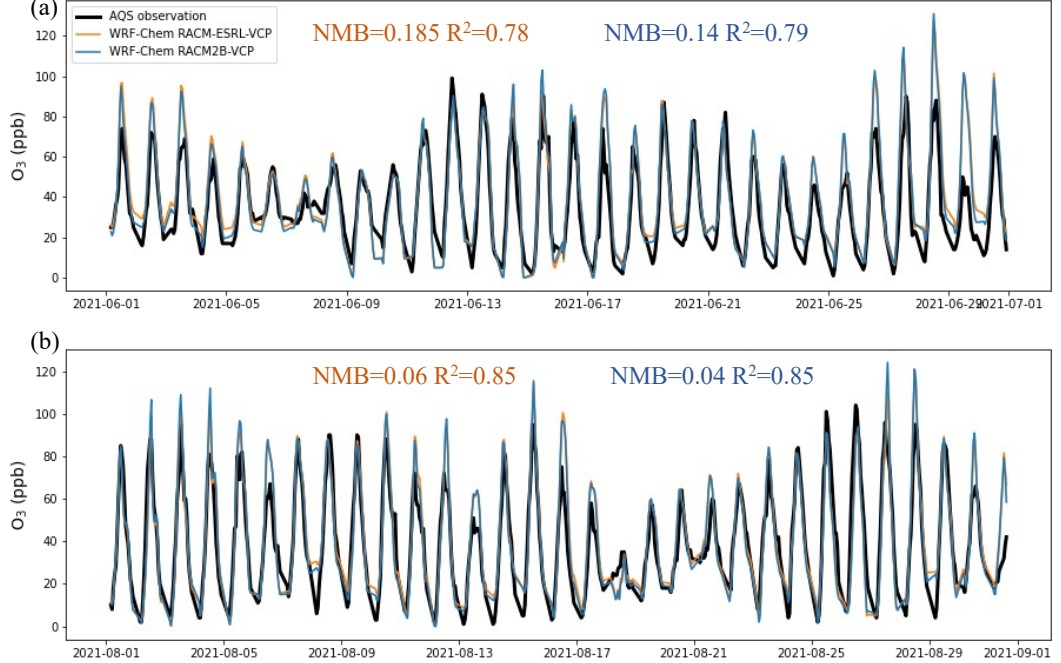

**Figure 2.** The comparison of time series of hourly $O_3$ between AQS site and two WRF-Chem simulations, one with RACM-ESRL-VCP and another one with RACM2B-VCP chemical mechanism at AQS site at Pasadena. The corresponding NMB and $R^2$ between model simulations and observations of hourly $O_3$ are shown on each plot.

Additionally, we analyzed the weekday-weekend difference in MDA8 $O_3$ and daily $PM_{2.5}$. In the LA Basin, $NO_x$ emissions on weekends substantially decline due to reduced heavy-duty truck activities compared to weekdays (McDonald et al., 2012; Kim et al., 2016) while VOC emissions stay relatively constant. Analyzing disparities between weekdays and weekends serves as a method to evaluate how $O_3$ and $PM_{2.5}$ pollution respond to a reduction in $NO_x$ emissions. The spatial distributions of the weekday-weekend difference of MDA8 $O_3$ and daily $PM_{2.5}$ from both observations and model simulations are shown in Figure

S5. In West/Central LA, the MDA8 $O_3$ is on average 15.0% (18.5%-9.5%, minimum-maximum) lower during weekdays. The East basin exhibits a 4.0% (6.0%-2.2%) lower weekday MDA8 $O_3$. In Figure 3(b), the model simulations are able to reproduce this weekday-weekend difference in MDA8 $O_3$. WRF-Chem with RACM-ESRL-VCP mechanism exhibits 7.8% (11.6%-2.5%) and 1.2% (5.5%-0.1%) lower MDA8 $O_3$ in the West/Central LA and the East basin; WRF-Chem with RACM2B-VCP mechanism exhibits 9.9% (12.5%-3.5%) and 1.1% (5.4%-0.2%) lower MDA8 $O_3$ in each region. For daily $PM_{2.5}$ in Figure

3(d), we observe an opposite weekday-weekend pattern between West/Central LA and East basin. The daily $PM_{2.5}$ is 6.9% (11.5%-2.0%) lower during weekdays in West/Central LA and is 8.8% (8.2%-9.5%) higher during weekdays in East basin. While both model simulations agreed well with the observations in West/Central LA (-6.6% for RACM-ESRL-VCP and -5.9% for RACM2B-VCP, only the model with the RACM2B-VCP mechanism could reproduce the positive weekday-weekend difference in the East basin, albeit still underestimating the magnitude (-2.2% for RACM-ESRL-VCP and 3.5% for RACM2B-



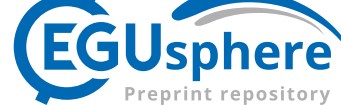

**Figure 3.** The comparison of MDA8 $O_3$ and daily $PM_{2.5}$ between AQS observations (gray) and WRF-Chem simulations configured with RACM-ESRL-VCP (orange) and RACM2B-VCP (blue) chemical mechanisms, respectively. (a) and (c) are the site-wise MDA8 $O_3$ and daily $PM_{2.5}$ averages; (b) and (d) are the weekday-weekend difference relative to the average of MDA8 $O_3$ ($\Delta$ MDA8 $O_3$) and daily $PM_{2.5}$ ($\Delta$ daily $PM_{2.5}$). The box is the interquartile range with the line of the median value and the black dot representing the mean value across the sites in West/Central LA and East Basin. The maximum and minimum values are shown by whiskers. The corresponding $R^2$ between model simulations and observations of MDA8 $O_3$ in West/Central LA and East basin is shown on each plot. While there are only 5 AQS sites with available $PM_{2.5}$ observations, the $R^2$ between model simulations and observations of daily $PM_{2.5}$ is calculated for all 5 sites regardless of the location.



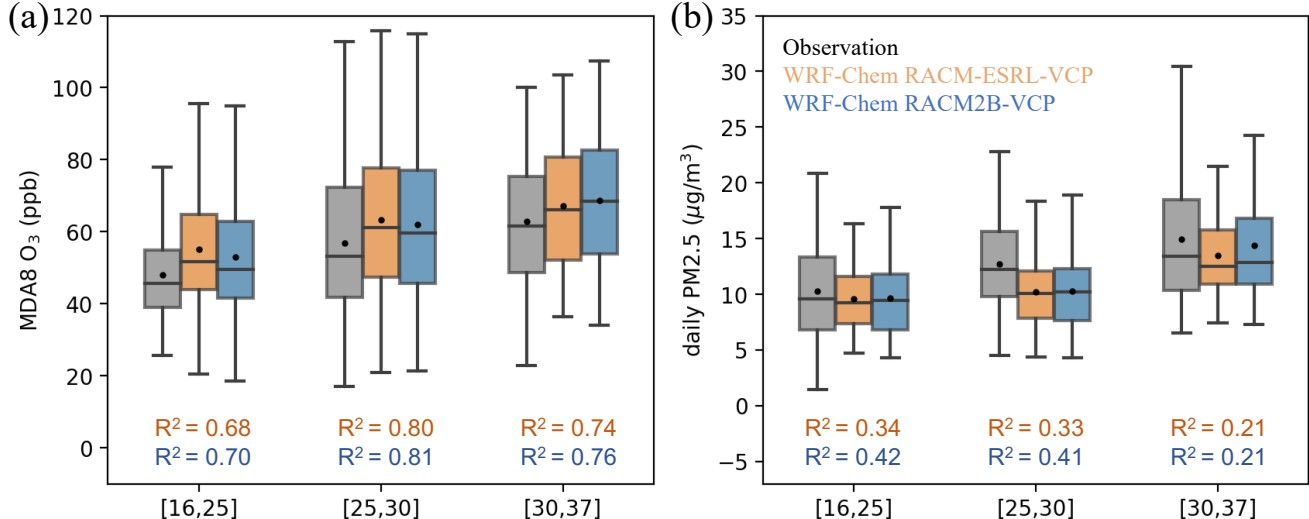

**Figure 4.** The comparison of the temperature dependence of MDA8 $O_3$ (a) and daily $PM_{2.5}$ (b) between AQS observations (gray) and WRF-Chem simulations configured with RACM-ESRL-VCP (orange) and RACM2B-VCP (blue) chemical mechanisms, respectively. MDA8 $O_3$ and daily $PM_{2.5}$ from observations and model simulations are segregated into days with three temperature bins, defined in Sect. 4. The variation in each bin is shown by a whisker plot; the black dot denotes the mean value and the line denotes the median value. The corresponding $R^2$ between model simulations and observations of MDA8 $O_3$ and daily $PM_{2.5}$ in each temperature bin is shown on each plot.

VCP). However, it is worth noting that this result might be biased due to the limited number of sites available for daily $PM_{2.5}$ measurements, only 3 sites in West/Central LA and 2 sites in East basin.

The model's ability to accurately simulate the temperature dependence of MDA8 $O_3$ and daily $PM_{2.5}$ is of utmost importance for accurately predicting pollutant trends in warmer climates. By segregating the data into three temperature bins defined in Sect. 4, we find significant positive temperature dependence of $O_3$ and $PM_{2.5}$ in both observations and model simulations,

depicted in Figure 4. The average MDA8 $O_3$ increases from 53.2 ppb (15.4 ppb, standard deviation) to 64.0 ppb (23.1 ppb) on days with low temperature to days with median temperature and further increases to 71.9 ppb (22.1 ppb) on days with high temperature. WRF-Chem simulations successfully replicate the temperature patterns of MDA8 $O_3$. A 9.6 ppb increase from low-temperature days to median-temperature days and a 6.2 ppb increase from median-temperature days to high-temperature days are found in WRF-Chem with RACM-ESRL-VCP mechanism; correspondingly, 10.6 ppb and 9.5 ppb increase are found

in WRF-Chem with RACM2B-VCP mechanism. However, for daily $PM_{2.5}$, the model failed to fully capture the temperature dependence, particularly exhibiting a low bias in median-to-high temperature days. While the observation presents an increase of 2.55 $\mu g/m^3$ in daily $PM_{2.5}$ from low-temperature days to median-temperature days, WRF-Chem simulations only see an increase of 0.63-0.68 $\mu g/m^3$ in daily $PM_{2.5}$.



In conclusion, the model simulations demonstrated promising results for both $O_3$ and $PM_{2.5}$ in the LA Basin. The RACM2B-
VCP mechanism performed as well as, and sometimes better than the RACM-ESRL-VCP mechanism, highlighting its potential
for air quality modeling in the region. Given the poor correlation between observation and model simulations in hourly $PM_{2.5}$
(Table S12) and the underprediction of $PM_{2.5}$ at higher temperatures (Figure 4), future efforts will prioritize improving the
representation of $PM_{2.5}$ in the RACM2B-VCP mechanism.

## 6  Evaluation of $NO_x$, CO and VOCs against RECAP-CA and SUNVEx field campaigns

Besides a direct comparison of $O_3$, it is important to verify whether the model accurately simulates ozone formation for the right
reason by evaluating modeled $O_3$ precursors against the observations. Compared to the RACM-ESRL-VCP mechanism, the
RACM2B-VCP mechanism offers a more complex representation of chemistry, making it more suitable for directly comparing
to observations in order to assess the chemistry accuracy.

To conduct this evaluation, we utilize airborne measurements from the RECAP-CA field campaign and surface measure-
ments from mobile drives during the SUNVEx field campaigns. Both flights and mobile drives sample the airmass covering
four regions of the LA basin, including the coastal region, Santa Ana Valley, Downtown LA, and San Bernardino Valley.
There is also a ground site at Pasadena, providing continuous ground site measurements during the SUNVEx field campaign.
MELODIES MONET, a new model evaluation tool (https://melodies-monet.readthedocs.io/), was used to pair the surface and
aircraft observations with the model results (Baker and Pan, 2017). To address the spatial variation, we match the WRF-Chem
simulation with the RACM2B-VCP chemical mechanism to the observations in time using linear interpolation and in space
using linear interpolation (vertical) and the nearest neighbor algorithm (horizontal), and the comparison between observations
and simulations is shown separately in these five regions. Instead of using NMB, we calculate the normalized median bias
(hereinafter referred to as NMDB, Eqn. 1) as the evaluation matrix.

$$NMDB = (\frac{\tilde{M}}{\tilde{O}} - 1) \tag{1}$$

Where $\tilde{M}$ and $\tilde{O}$ are the median of the model simulation and the observations, respectively. We use NMDB as the mobile and
ground measurements can capture sharp gradients of trace gases as they pass by large emission sources, which cannot be re-
solved by the spatial resolution of the model (4km). We note that NMDB reduces the influence of large local emission sources
as compared to NMB, but the impact of local sources cannot be completely removed using NMDB. Table 1 summarizes the
NMDB and the $R^2$ of our studied species from the WRF-Chem simulation with the RACM2B-VCP chemical mechanism,
including $NO_x$, CO, VOCs, PAN, and aerosols, comparing against three different observations. Overall, we show that the $R^2$
is generally higher between model simulation and airborne measurements. The $R^2$ is the lowest compared to mobile measure-
ments, suggesting that the model is too coarse to completely resolve the spatial variation observed in mobile drives, which by
design are prone to be impacted by local emission sources.



| Species | RECAP airborne | SUNVEx mobile | SUNVEx ground |
|---|---|---|---|
| $NO_x$ | -0.29 (0.55) | N/A | 0.29 (0.18) |
| CO | -0.20 (0.68) | N/A | -0.10 (0.10) |
| Calibrated $VOC_r$ | -0.35 (0.58) | -0.26 (0.03) | -0.04 (0.28) |
| Species-breakdown calibrated $VOC_r$ | -0.36 | -0.21 | -0.21 |
| D5-Siloxane | 0.59 (0.56) | 1.18 (0.26) | 1.67 (0.61) |
| PCBTF | -0.05 (0.32) | 0.43 (0.004) | 1.19 (0.13) |
| $CH_4$ | -0.04 (0.48) | -0.04 (0.003) | -0.03 (0.25) |
| Methanol | 0.01 (0.67) | 0.04 (0.08) | 0.42 (0.04) |
| Ethanol | -0.75 (0.52) | -0.41 (0.08) | -0.30 (0.33) |
| Acetaldeyhyde | 0.47 (0.29) | -0.48 (0.03) | -0.18 (0.60) |
| Acetone | -0.33 (0.65) | -0.14 (0.02) | 0.02 (0.24) |
| Isoprene | -0.46 (0.25) | -0.01 (0.18) | -0.23 (0.60) |
| MACR+MVK | -0.48 (0.61) | 0.11 (0.52) | -0.28 (0.64) |
| Monoterpene | -0.89 (0.20) | -0.30 (0.01) | -0.51 (0.06) |
| Benzene | 0.06 (0.63) | 0.46 (0.03) | 1.49 (0.30) |
| Toluene | -0.05 (0.49) | -0.12 (0.01) | 0.76 (0.17) |
| Benzaldehyde | 0.05 (0.34) | 1.18 (0.02) | 3.00 (0.18) |
| Xylene | -0.09 (0.53) | 0.23 (0.01) | 1.35 (0.18) |
| PAN | N/A | N/A | -0.73 (0.55) |
| Organic aerosol | N/A | N/A | 0.03 (0.16) |
| Sulfate aerosol | N/A | N/A | 0.09 (0.05) |
| Ammonia aerosol | N/A | N/A | 0.15 (0.17) |

**Table 1.** Summary of normalized median bias (NMDB) and the $R^2$ (in the parenthesis) for $NO_x$, CO, calibrated VOCs species and $VOC_r$, PAN and speciated aerosols from WRF-Chem simulation configured with RACM2B-VCP mechanism, comparing against three field campaigns, including RECAP airborne measurements, SUNVEx mobile measurements, and SUNVEx ground measurements. The comparisons of PAN and aerosols are only available against SUNVEx ground measurements. The calibrated $VOC_r$ is the sum of $VOC_r$ from each VOC species, matching each observation point. We also first calculate the median $VOC_r$ from each VOC species during the field campaign and then sum them up to the "species-breakdown calibrated $VOC_r$", to avoid the compensating effect across multiple VOCs.





**Figure 5.** The comparison of $NO_x$ (a), CO (b), D5-Siloxane (c), PCBTF (d) and calibrated $VOC_r$ between observations and WRF-Chem simulation configured with RACM2B-VCP chemical mechanism. The comparison to the RECAP airborne measurement in yellow shade; SUNVEx mobile measurements in red shade and SUNVEx ground measurements in green shade. The distribution is shown by a whisker plot; the black dot denotes the mean value and the line denotes the median value.





We compare the distribution of trace gases in each of the five regions (Coastal region of LA, Santa Ana Valley, Downtown
LA, San Bernardino Valley, and Pasadena) from the model simulation with the RACM2B-VCP mechanism against the obser-
vations. Figure 5(a) and (b) present the evaluation of $NO_x$ and CO. We exclude the measurements of $NO_x$ and CO from the
mobile drives as they are too near to traffic sources that cannot be captured by the model simulation at 4km. The observed $NO_x$
exhibits a wide spatial variation, which is consistent with the spatial pattern of CO. Downtown and the San Bernardino Valley
are characterized by higher $NO_x$ concentrations with a median of 5.2 ppb, which is due to its large urban population, as well
as meteorological and geographic conditions favorable for pollution accumulation during the day. The coastal region in LA
has the lowest $NO_x$ concentrations, a median of 2.4 ppb, as it experiences sea breeze and features lower local $NO_x$ emissions
(Nussbaumer et al., 2023). The WRF-Chem model with the RACM2B-VCP mechanism shows reasonable agreement with
observed $NO_x$ and it varies by region. The best agreement is found at Santa Ana Valley with an NMDB of -4.9%. The model
overpredicts $NO_x$ by an NMDB of 30% in Pasadena while it underpredicts $NO_x$ in Downtown LA and the San Bernardino
Valley by an NMDB of -31.1% and -31.8%. These biases can be, at least partially, explained by the fact that flights during RE-
CAP flew near or above the highways most of the time in these two regions so the observations can be biased high compared to
the 4km grid averages. A better agreement is found for the $NO_x$ vertical profiles between WRF-Chem simulations at 1km and
observations during RECAP (Yu et al., 2023, in review). In the meanwhile, new mega warehouses in the San Bernardino Valley
are adding truck traffic that is not considered in the inventory (Nussbaumer et al., 2023). The model simulation underpredicts
CO concentrations in all regions while still reproducing the spatial pattern. The NMDB is -20% for all observations, with the
largest underprediction with the NMDB of -26.4% in Downtown LA and the lowest underprediction with the NMDB of -10.1%
in Pasadena.

The inclusion of the VCP tracers in the RACM2B-VCP mechanism allows us to compare model-predicted vs observed
concentrations of VCP markers and provide insight into VCP emissions. Two VCP tracers, D5-Siloxane and PCBTF, are
calibrated and reported from RECAP and SUNVEx field campaigns. Shown in Figure 5(c) and (d), we find that the WRF-
Chem simulation with RACM2B-VCP mechanism agrees well with airborne measurements for both D5-siloxane and PCBTF,
featuring the NMDBs of 59.4% and -5.4%. Larger biases are found between the model and measurements near the surface as
the corresponding NMDBs are 144% for D5-siloxane and 72.8% for PCBTF. The largest bias occurs in Downtown LA where
the model overpredicts the median D5-siloxane by a factor of 2.

Next, we evaluate the VOC chemistry thoroughly using a set of calibrated VOCs from both RECAP and SUNVEx cam-
paigns. These VOCs are either emitted from a wide range of sources or from secondary production. In addition to D5-Siloxane
and PCBTF emitted from VCP sources, aromatics, including benzene, toluene, xylene, and benzaldehyde, are primary VOCs
predominantly emitted from mobile sources. Isoprene, and its oxidative products, methacrolein (MACR) and methyl vinyl
ketone (MVK) are solely from biogenic emissions. Monoterpenes are emitted from a mixture of biogenic sources and VCP
products. We also measure a series of oxygenated VOCs, including methanol, ethanol, acetone, and acetaldehyde. The VOC
reactivity is calculated using the concentrations, from either observations or the model simulation, times the OH reaction rate
from the RACM2B-VCP chemical mechanism for each calibrated species, and the comparison of total calibrated VOC reac-
tivity is shown in Figure 5(e). In the absence of alkanes, alkenes, and formaldehyde, the calibrated VOC reactivity comprises





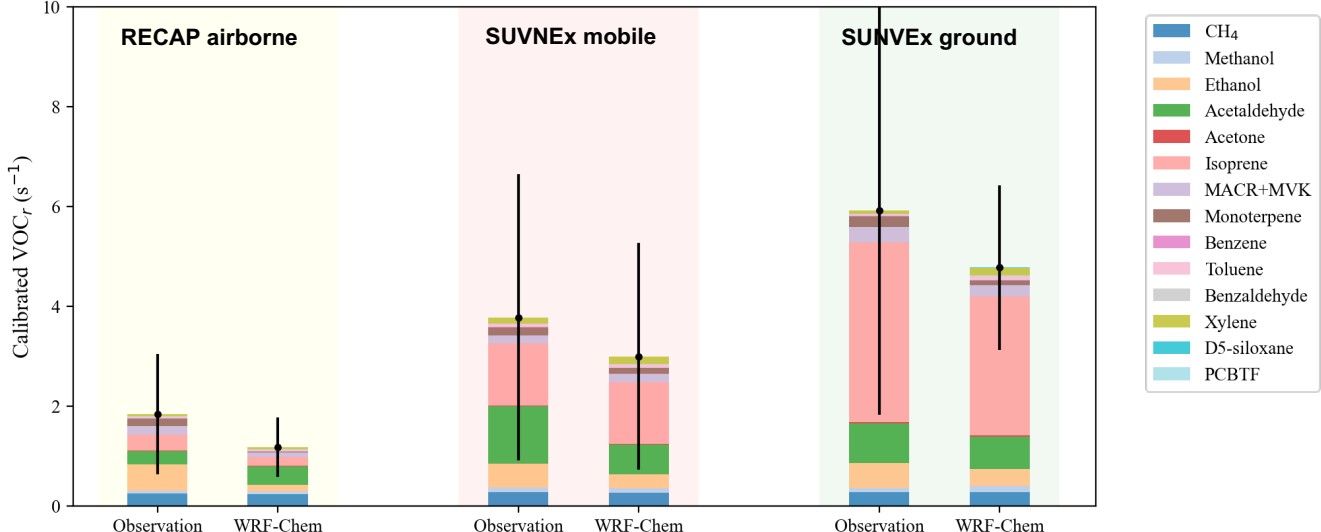

**Figure 6.** The comparison of median VOC reactivity (VOC$_r$) from each calibrated species between observations and WRF-Chem simulation configured with RACM2B-VCP chemical mechanism. The comparison to the RECAP airborne measurement in yellow shade; SUNVEx mobile measurements in red shade and SUNVEx ground measurements in green shade. The black line indicates the standard deviation of summed calibrated VOC$_r$ in either observations or model simulations.

58% of the total VOC reactivity in WRF-Chem (Figure S6). We find that the calibrated VOC reactivity is underrepresented (Figure 5(e)); The NMDB is -34.9% compared to the airborne measurements, ranging from -42.9% in Downtown to -32.5% in Santa Ana Valley. The NMDB is -17.1% compared to the mobile measurements, ranging from -48.2% in Santa Ana Valley to -17% in the Coastal region. The modeled calibrated VOC reactivity is closer to the ground observations at Pasadena, with an NMDB of -4.8%.

The underprediction of VOC reactivity becomes more pronounced and consistently apparent across the three sets of observations when assessing the individual contribution from VOC species. In Figure 6, we first calculate the median VOC reactivity from each calibrated species, either from observations or from the model simulation, and then sum it up to the total calibrated VOC reactivity. The NMDB of summed calibrated VOC reactivity is -36.2% for the airborne measurements, and -20.1% for both mobile and ground measurements. The primary VOCs from traffic sources, benzene for instance, show good agreement against airborne and mobile measurements; the NMDB is -4.8% for the airborne measurements and 18.8% for the mobile measurements across the basin. However, WRF-Chem turns out to overpredict benzene by an NMDB of 128% at the ground site (Figure S7(c)). In other words, the measurement at one ground site at Pasadena alone is not representative of the LA basin overall. Pasadena stands out due to abundant biogenic sources, which is reflected in a 65% contribution of total VOC reactivity from isoprene and its oxidative products. At Pasadena, the model underpredicts the isoprene by an NMDB of -24.1% and also underpredicts the MACR+MVK by an NMDB of -31.4%. We note consistent underprediction of monoterpene and ethanol in WRF-Chem; the largest low bias in the model is found when comparing against the airborne measurements, featuring an




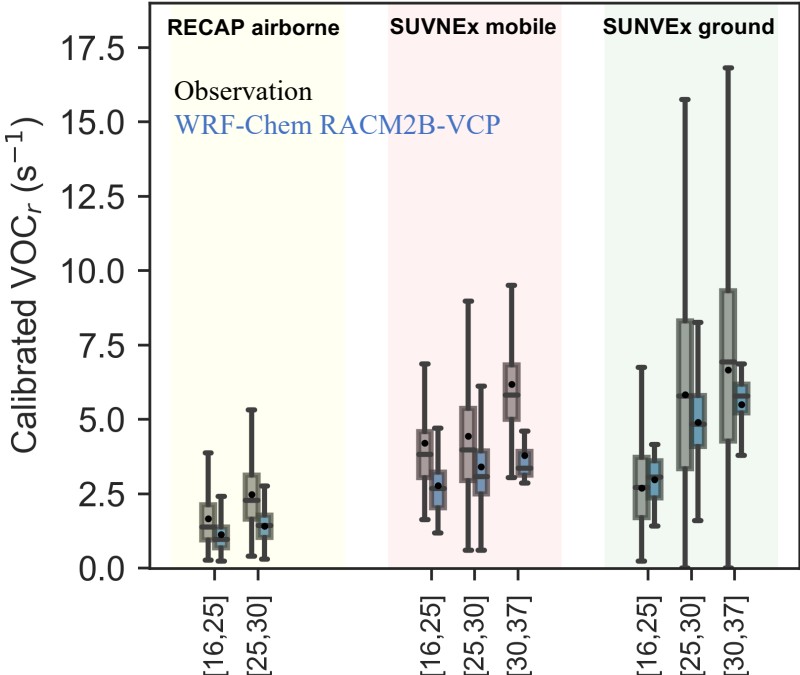

**Figure 7.** The comparison of the temperature dependence of VOC reactivity (VOC$_r$) between observations (gray) and WRF-Chem simulation configured with RACM2B-VCP chemical mechanism (blue). The comparison to the RECAP airborne measurement in yellow shade; SUVNEx mobile measurements in red shade and SUVNEx ground site measurements in green shade. The distribution is shown by a whisker plot; the black dot denotes the mean value and the line denotes the median value.

NMDB of -89.5% for monoterpene and -75.4% for ethanol. The bias is smaller near the surface as the NMDB is -45.1% for monoterpene and -35.4% for ethanol. Acetaldehyde also contributes to the low bias in VOC reactivity in the model. The NMDB is -48.3% and -22.9% for measurements from mobile drives and from the ground site at Pasadena. However, the model overpredicts acetaldehyde when compared to the airborne measurements characterized by an NMDB of 46.4%.

We also explore the temperature dependence of the total calibrated VOC reactivity, as depicted in Figure 7. Strong positive temperature dependence of the calibrated VOC reactivity is evident across the three sets of observations. Although the nine flight days during RECAP do not cover days with defined high temperatures, the calibrated VOC reactivity shows an increase from 1.39 s$^{-1}$ to 2.27 s$^{-1}$ from days with low temperatures to median temperatures. Both mobile and ground measurements cover days with three different temperature bins. The calibrated VOC reactivity increases from 3.83 s$^{-1}$ to 5.82 s$^{-1}$ in mobile

measurements and from 2.7 s$^{-1}$ to 6.9 s$^{-1}$ in ground site measurements. It is promising to see that the model reproduces this temperature dependence despite underpredicting the calibrated VOC reactivity.

Furthermore, Figure S7 illustrates the temperature dependence of some key VOC species. Strong temperature dependence is evident in both observations and model simulations for species like isoprene, primarily due to temperature-dependent emis-





sions. Conversely, for species emitted from traffic and VCP emissions, no consistent temperature dependence is found in either
the observations or the model simulation. However, it is worth noting that temperature dependence is not solely attributed to
emissions. For instance, D5-siloxane, PCBTF, and benzene are emitted from sources that are temperature-neutral in our model
simulation, yet a positive temperature dependence exists in Pasadena, driven by meteorological factors.

## 7 Evaluation of PAN and aerosols against ground site measurements from the SUNVEx field campaigns

In Section 6, we conducted a model evaluation of trace gases observed and reported in three sets of observations. Additional
measurements of peroxyacetyl nitrate (PAN) and aerosols are available in the ground site measurements. While PAN is treated
as a surrogate species in the RACM-ESRL-VCP mechanism, the RACM2B-VCP mechanism separates it out from the higher
carbon acyl nitrates such as peroxypropionyl nitrate (PPN), enabling a direct comparison to the observations. As shown in
Figure 8(a), the model with the RACM2B-VCP mechanism underestimates PAN levels at Pasadena. The median PAN concen-
tration is 1.81 ppb in observations and 0.48 ppb in the model, resulting in an NMDB of -72.7%.

We compare the aerosols in Figure 8(b), categorized into four groups: organic aerosols (OAs), sulfate aerosol, ammonium
aerosol, and nitrate aerosol. OAs comprise the majority (78%) of the aerosols observed at ground site measurements, followed
by sulfate aerosol (12%), ammonium aerosol (5.6%), and nitrate aerosol (4.4%). When compared to the observations, the model
with the RACM2B-VCP mechanism yields excellent agreements for aerosols in the first three categories, with an NMDB of
2.9% for OAs, 15.6% for sulfate aerosol, and 9.4% for ammonium aerosol. However, the model fails to simulate nitrate aerosols.
The model also reproduces the temperature dependence of OAs, as shown in Figure S8.

## 8 The impact of VCP emissions on VOC reactivity and $O_3$

As we have fully evaluated the VOC emission and VOC chemistry from each emission sector, including fossil fuel, VCP,
and biogenic sources, in this section, we will explore the influence of VCP emissions on VOC reactivity and $O_3$. We do not
include $PM_{2.5}$ here as the model needs to be improved to better represent the speciation for SOA formation (Section 2.3),
the hourly variation in $PM_{2.5}$ (Section 5), and the daily $PM_{2.5}$ temperature dependence (Section 5) before it can be used for
source apportionment analysis. In addition to the aforementioned model simulations (referred to as $S_0$), we conducted two
distinct sensitivity tests in August 2021 to gain further insights and quantify the impacts of VCP sources. In the first sensitivity
simulation ($S_1$), we only exclude the VCP source; in the second sensitivity simulation ($S_2$), we exclude all anthropogenic
VOC emissions. The vast majority of the anthropogenic emissions outside of VCP are fossil fuel VOC emissions, either from
combustion or evaporation and they will be referred to as fossil fuel VOCs from here on. The disparity between $S_1$ and $S_0$
delineates the extent to which VCP sources contribute, whereas the disparity between $S_2$ and $S_1$ elucidates the impact of fossil
fuel VOC emissions in August 2021.

As depicted in Figure 9(a), anthropogenic VOC emissions account for 46%-56% of the total VOC reactivity across the LA
region. Notably, VCP sources constitute 36%-63% of the anthropogenic influence. The relative contribution of VCP emissions



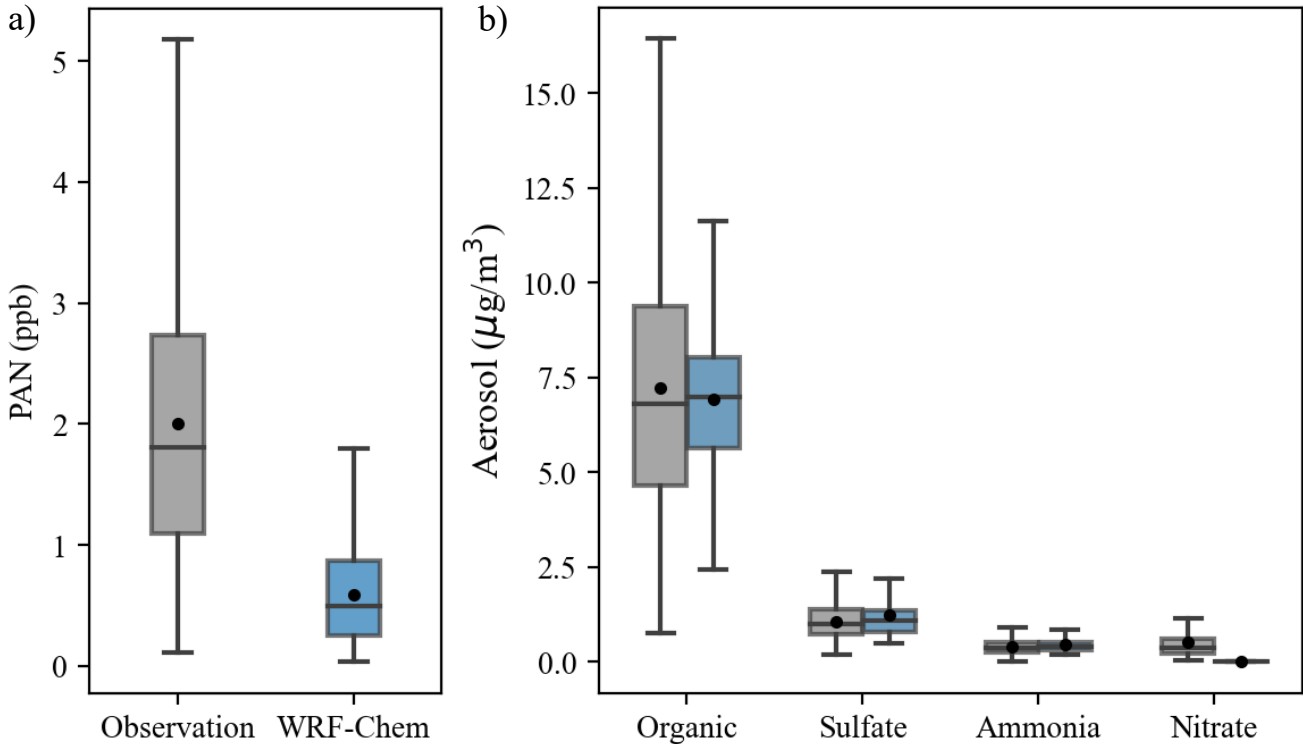

**Figure 8.** The comparison of PAN (a) and speciated aerosols (b) between ground observations at Pasadena (gray) and WRF-Chem simulation with RACM2B-VCP mechanism (blue). The distribution is shown by a whisker plot; the black dot denotes the mean value and the line denotes the median value.

to the total VOC reactivity varies by region, spanning from 21% along the coastal areas of LA to 35.5% in Downtown LA. Figure 9(b) shows the contribution of MDA8 ozone from these two anthropogenic VOC emissions. In addition, in our model simulation, the LA basin experiences the background $O_3$ of 30 ppb transport outside of the model domain defined by the boundary condition from RAQMS. We show a larger contribution of local $O_3$ production moving from the coastal regions to the east, which is consistent with the positive $O_3$ gradient as we have observed in Figure 3(a). Compared to the contribution to

the total VOC reactivity, the contribution of both VCP and fossil fuel VOC emissions to MDA8 $O_3$ are smaller. The smallest VOC emission contributions are observed along the coastal areas of LA, totaling 5.0%, with VCP emissions responsible for 1.6% of the MDA8 $O_3$ enhancement. In contrast, San Bernadino exhibits the most substantial contribution from anthropogenic VOC emissions at 21.6% in total, with VCP emissions responsible for 10.6% of the MDA8 $O_3$. Our findings align with the results of Coggon et al. (2021), where they ascertain that out of the 43 ppb of $O_3$ attributed to VOC emissions in New York City,

roughly half stems from anthropogenic sources. Within this anthropogenic fraction, over 50% is attributable to VCP emissions. However, this study reports a lower contribution of VCPs to MDA8 $O_3$ than Qin et al. (2021) where they showed that VCPs produce 17% of MDA8 $O_3$ ($9 \pm 2$ ppb) in summer 2010 in Los Angeles.



While we have demonstrated the capability of our WRF-Chem simulation with the RACM2B-VCP mechanism to capture the temperature dependence of VOC reactivity and $O_3$, we conducted a more detailed investigation into the contributions of the two anthropogenic VOC emissions to VOC reactivity and $O_3$ under various temperature conditions. Instead of segregating the study domain into four regions as shown in Figure 9, we addressed spatial differences by dividing the study domain into West/Central LA and the East basin, as described in Sect. 5. Within each region, we calculated the relative contributions from these anthropogenic VOC emissions in three temperature bins to the total VOC reactivity and MDA8 $O_3$ (Figure 10 and Figure S9(a, b)). Our findings indicate that the relative contribution of anthropogenic VOC emissions to VOC reactivity decreases on hotter days, primarily due to the temperature sensitivity of VOCs emitted from biogenic sources, which increase at higher temperatures. This reduction in relative contribution to VOC reactivity is notably more pronounced in the East basin, where biogenic sources are more abundant. The fractional ratio of VOCs from VCP and fossil fuels decreases by 1.7% and 2.1%, respectively.

However, we highlight that despite this reduction in VOC reactivity, the relative contribution of VOCs from these two anthropogenic emissions to MDA8 $O_3$ increases at elevated temperatures. Specifically, the relative contribution of anthropogenic VOC emissions increases from 7.8% to 16.4% across the lowest to the highest temperature bins in West/Central LA and from 10.6% to 17.5% in the East basin. VOCs from both VCP and fossil fuel emissions contribute equally to MDA8 $O_3$ formation. This increasing contribution from anthropogenic VOC emissions is also evident when only considering the local enhancement of MDA8 $O_3$ by subtracting background transport outside of the model domain (Figure S10). The diverging trends observed in total VOC reactivity and MDA8 $O_3$ emphasize the complexity of changes involving the chemical regimes.

In addition to total VOC reactivity and MDA8 $O_3$, we observed a strong temperature dependence in surface $NO_x$ over LA, influenced by meteorological factors. Despite $NO_x$ emissions in LA being predominantly anthropogenic and not responsive to temperature changes in our emission inventory, hotter days tend to induce stagnation and weaker mixing, resulting in higher $NO_x$ levels at the surface. Given the entangled effects of both emissions and meteorology on $NO_x$ and VOC reactivity under different temperatures, directly mapping the relative changes in VOC reactivity to the relative contribution to $O_3$ formation proves challenging. The combined effect of emission sources and meteorological conditions results in a larger contribution to $O_3$ formation due to VOCs emitted from VCP and fossil fuel sources over LA at higher temperatures.

## 9 Conclusion

We have developed the novel chemical mechanism RACM2B-VCP for WRF-Chem, aimed at enhancing the representation of VOC chemistry in present urban areas where mobile emissions are declining and emissions from other sources like biogenics and VCPs are contributing more to VOC reactivity and ozone formation. RACM2B-VCP, an extension of the RACM2-Berkeley2 mechanism, incorporates oxygenated VOC reactions to address the influence of VCPs. The integration of the TUV photolysis scheme, a more complex SOA VBS scheme, and aerosol uptake reactions further refine the representation of photolysis, aerosols, and ozone, respectively. Additionally, we have introduced VCP tracers, D4 siloxane, D5 siloxane, *p*-Dichlorobenzene, and PCBTF, into RACM2B-VCP.





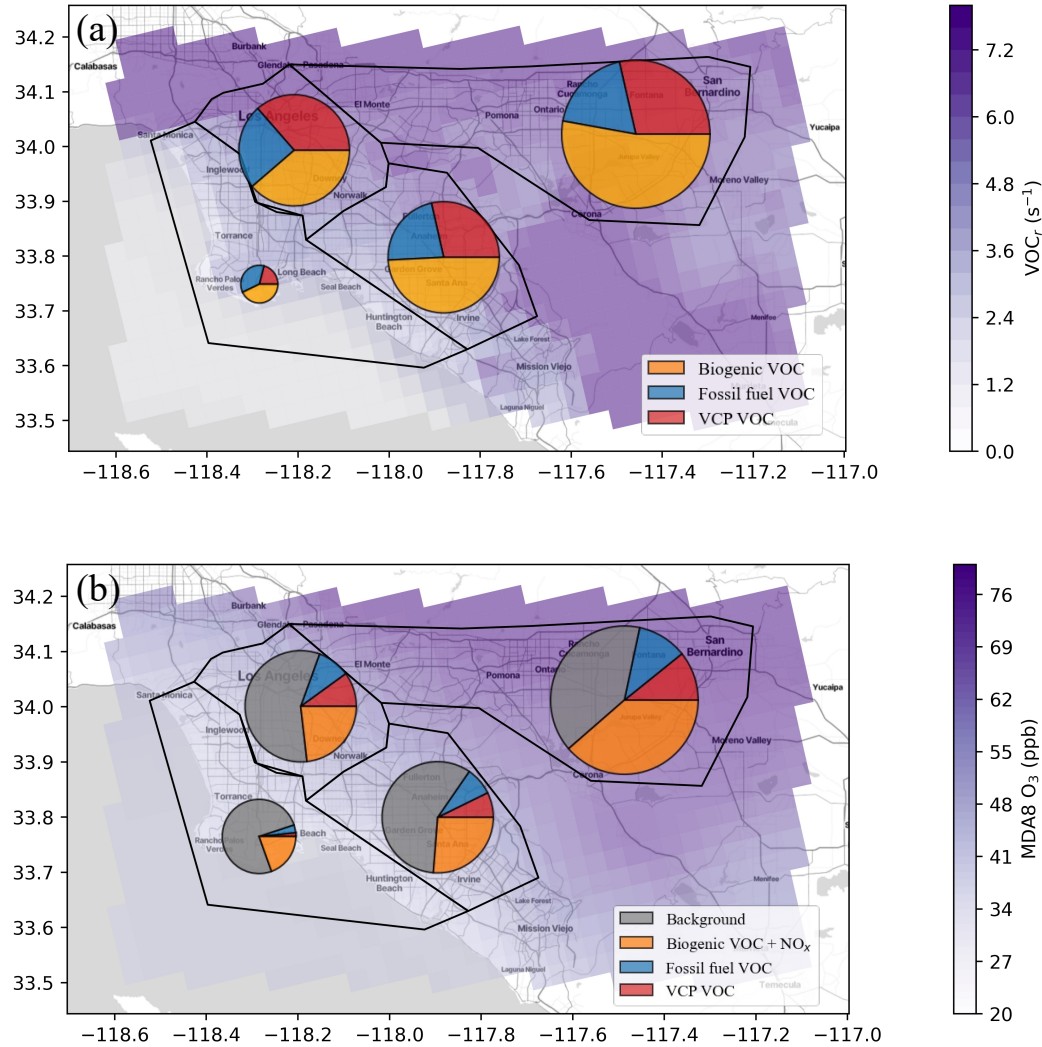

**Figure 9.** Spatial distribution as well as the budget analysis of (a) total VOC reactivity and (b) MDA8 O$_3$ averaged in August 2021. a) shows the contributions of VOC emissions from VCP and fossil fuel sources to total VOC reactivity over four regions of LA, including coastal LA, downtown LA, Santa Ana Valley, and San Bernadino Valley. The total VOC reactivity is the average of 9 am to 8 pm local time. b) shows the contribution of MDA8 O$_3$ from VCP, fossil fuel VOC emissions as well as background O$_3$ (ppb).



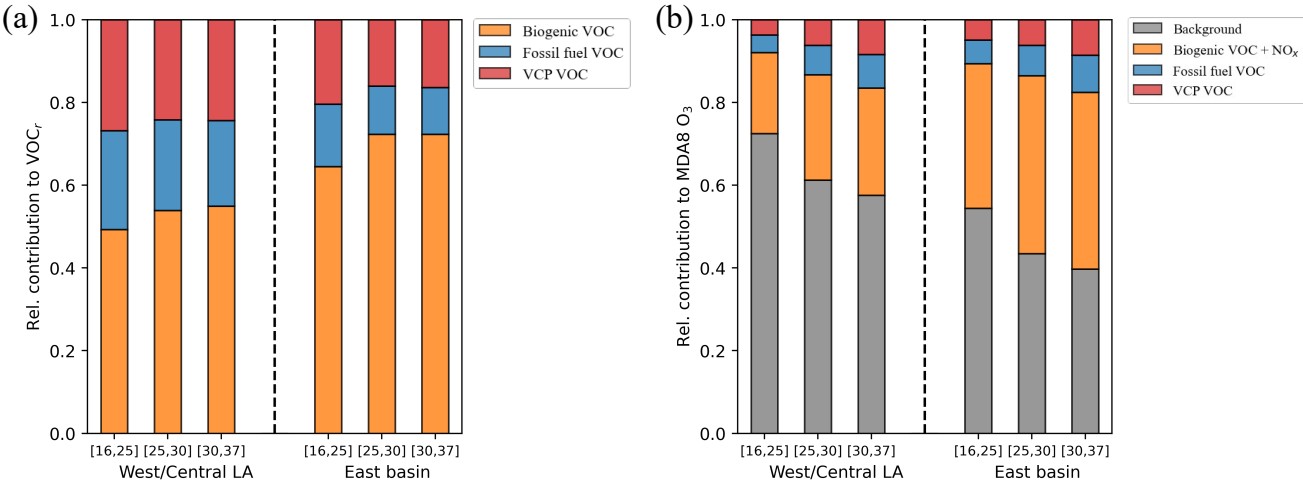

**Figure 10.** The relative contribution of VOC emissions from VCP and fossil fuel sources to VOC reactivity (a) and MDA8 $O_3$ (b), in West/Central LA and in East basin, under three temperature bins defined in Figure 1.

We evaluate the performance of RACM2B-VCP by comparing WRF-Chem simulations with AQS surface network data and the previous RACM-ESRL-VCP mechanism. Notably, RACM2B-VCP exhibits comparable, if not superior depending on the metric, performance in reproducing MDA8 $O_3$ and daily $PM_{2.5}$, capturing both its magnitude and spatial distribution. While the model fails to capture the hourly variation in $PM_{2.5}$, future work is needed to provide better constraints on SOA such as what might be accomplished through coupling gas and aerosol-phase chemistry as in CRACMM (Pye et al., 2023). We also conduct a comprehensive assessment of the RACM2B-VCP mechanism by comparing against measurements from RECAP and SUNVEx field campaigns. While our model adeptly represents CO, $NO_x$, and VOCs from various sectors, there remains a disparity in total measured VOC reactivity against observations. The most underpredicted VOC species are monoterpene, ethanol, and acetaldehyde, which suggest likely missing sources of VOCs.

Our study also demonstrates the notable impact of VCP emissions on both total VOC reactivity and MDA8 $O_3$ in August 2021 averaged across the LA basin. VCP sources contribute significantly, accounting for 31% of total VOC reactivity and contributing 6.4% to MDA8 $O_3$. It underscores the importance of considering VCP emissions in urban air quality management strategies. We also evaluate the relative contributions of VOCs from VCP and fossil fuel emissions to total VOC reactivity and MDA8 $O_3$ across different temperatures. While the contribution of anthropogenic VOC emissions decreases for total VOC reactivity, both VCP and fossil fuel VOC emissions show a larger contribution to MDA8 $O_3$ at higher temperatures, suggesting non-linear changes in the $O_3$ formation due to changes in the chemical regime.

Looking ahead, our study suggests a roadmap for further refining VOC chemistry simulations:

1. Updating the biogenic emission inventory, particularly over urban regions.

2. Implementing a more accurate SOA scheme to better characterize the source apportionment of $PM_{2.5}$.



3. Addressing the existing gaps between observational VOC reactivity and model predictions, thereby improving the overall model representation.

*Code and data availability.* The observational data from SUNVEx and RECAP field campaigns and WRF-Chem anthropogenic emission files are available https://csl.noaa.gov/projects/sunvex/. The analysis dataset is available at https://csl.noaa.gov/groups/csl4/modeldata/ data/Zhu2023/. The WRF-Chem source codes and the analysis codes are available at https://github.com/NOAA-CSL/WRF-Chem_CSL_ 515    Publications/tree/main/Qindan_Zhu_et_al_2024.

*Author contributions.* RHS and BM supervised the research; QZ performed the analysis and prepared the manuscript; QZ conducted the model simulations with the help of RHS, BM, CH, JS, JH, HOTP, ML, BBm ZM, RA, EP, BP, PW, BS, CA, AB, JS, AHG, and RCC conducted the RECAP field campaign and provided the measurements; MMC, CW, CES, LX, KZ, MAR, AN, PRV, JP, SSB are involved in SUNVEx field campaign and provided the measurements; all authors have reviewed and edited the paper.

*Competing interests.* The authors have the following competing interests: At least one of the (co-)authors is a member of the editorial board of Atmospheric Chemistry and Physics.

*Acknowledgements.* QZ was supported by funding from the EPA STAR program and the NOAA Climate & Global Change Postdoc Fellowship. MMC, CES, QZ, and RHS received support from the U.S. Environmental Protection Agency (EPA) STAR program (grant # 84001001). The views expressed in this document are solely those of the authors and do not necessarily reflect those of the Agency. EPA does not endorse 525    any products or commercial services mentioned in this publication. This RECAP field campaign was funded by California Air Resources Board Contract number 20RD003, 20AQP012, and the Presidential Early Career Award for Scientists and Engineers (PECASE) (from Brian McDonald). This work was partly supported by NOAA Cooperative Agreement with CIRES, NA17OAR4320101 and NA22OAR4320151.



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
