# Peer review of "A better representation of VOC chemistry in WRF-Chem and its impact on ozone over Los Angeles"

_EGUsphere, 2023_

## Referee Comment (RC1)

**General comments**
The authors develop a new chemical mechanism, RACM2B-VCP for WRF-Chem that can better represent VCP sources in the urban environment. They evaluate their model against observations from the RECAP-CA airborne campaign and the SUNVEx ground and mobile lab campaign. The authors find that 52% of the VOC reactivity and 35% of the local enhancement of MDA8 ozone come from anthropogenic VOC emissions, and 50% of this is attributable to VCP emissions. This manuscript is a helpful description of model implementation and analysis of VCP emissions and chemistry and should be accepted for publication after minor revisions described below.

The authors discuss the difficulty in comparing absolute concentrations from WRF-Chem to ground site measurements. Could the authors use emission ratios or other chemical coordinates to better take advantage of these measurements and use them to constrain the performance of the VCP mechanism? The authors describe adding 8 VOCs to the model. The paper would be more beneficial to policy-makers and other modelers if they could discuss which VCPs are most contributing to ozone production. Is ethanol the key player? Or are there significant contributions from other species?

 **Specific Comments**
Line 35 – The statement that VCPs are an 'emerging source' could be read to mean that VCPs are a new source. Is that what you mean? It is my understanding that VCPs have always existed? Or do you mean 'recently recognized' or something similar?

Line 50 – Can you explain why this is: "reduced NOx results in higher ozone at low temperatures"?

Supplement – The chemistry for eucalyptol is not listed in the supplement.

Line 189 – What is the vertical resolution of the model?

Line 209 – Are the emissions monthly? Do you apply diurnal or weekend/weekday scale factors to the emissions?

Line 247 – Can you explain what you mean by micro-scale and middle-scale environments? Are these sites near am emissions source? Or do you mean to imply something about the geography?

Figure 2 – Can you clarify which site is in Figure 2?

Figure 2 – It looks like WRF-Chem RACM-ESRL-VCP has higher nighttime ozone than WRF-Chem RACM2B-VCP. If so, could you explain why this might be? Do you see a reduction in model nitrate aerosol formation that improves comparisons with observations?

Line 373 – The bias of 59% for D5-siloxane is high to be called 'good agreement'. Can you comment more on the possible reasons for this bias and what it might imply about the VCP inventory?

Line 377 – Toluene and C8 aromatics can have a source from solvents (https://doi.org/10.5194/acp-21-6005-2021). Can you look at a weekend/weekday analysis for species you attribute to traffic or does changes in their lifetime make this too difficult?

Line 378 – I have heard some discussion of anthropogenic sources of isoprene. Have you looked at whether isoprene correlates with any anthropogenic tracers to support the statement that it is solely biogenic?

Figure 6 – It looks like the model overpredicts acetaldehyde even though it underpredicts ethanol, a major precursor, by a large amount. Is there a large primary source of acetaldehyde in the model?

Line 414 – Should some VCPs that evaporate have a temperature dependence (such as solvents) or is this effect small? Could you discuss expectations for temperature dependence of VCPs a little more here?

Line 424 – Here could you look at PAN vs. acetaldehyde to look at whether the chemistry is producing PAN at the correct rate even if it is underestimated due to model resolution etc?

Line 429 – Can you provide any reason for the model underestimate of nitrate aerosol, particularly given that the model correctly simulates sulfate and ammonium? Could the model NOx be too low? Or the model RH be incorrect etc? Why does the model both underestimate nitrate but also simulate negligible nitrate variability?

Line 482 – Can you discuss the implications of the model underestimate in VOC reactivity (Fig. 6 & 7) but the overestimate in D5-siloxane on your findings in this section? What further constraints are needed to improve the inventory or the model? For example, do you think the large monoterpene underestimate is biogenic or from VCPs? Could this be a good place to discuss D5-siloxane in ratios to other species emitted from adhesives and personal care products? Or are VCP species like ethanol spread across too many sources to perform this type of analysis?

Line 499 – What do you think the cause of these model underpredictions of monoterpenes, ethanol, and acetaldehyde could be?

Line 508 – The statement "Updating the biogenic emission inventory, particularly over urban regions." in the conclusions is unclear. It would be better if the authors wrote out the reasoning behind each bullet point. For example, "Our study shows we need to update the biogenic emission inventory because of model underestimates in isoprene and monoterpenes"?

Figure S9 – How do VOCs contribute to NOx?

---

## Author Comment (AC1)

**Response to reviewers "A better representation of VOC chemistry in WRF-Chem and its impact on ozone over Los Angeles"**

Qindan Zhu[1,2,*], Rebecca H. Schwantes[1], Matthew Coggon[1], Colin Harkins[1,2], Jordan Schnell[1,2], Jian He[1,2], Havala O. T. Pye[3], Meng Li[1,2], Barry Baker[13], Zachary Moon[13,14], Ravan Ahmadov[4], Eva Y. Pfannerstill[5], Bryan Place[3], Paul Wooldridge[6], Benjamin C. Schulze[7], Caleb Arata[8], Anthony Bucholtz[9], John H. Seinfeld[10], Carsten Warneke[1], Chelsea E. Stockwell[1,2], Lu Xu[1,2,***], Kristen Zuraski[1,2], Michael A. Robinson[1,2], Andy Neuman[1], Patrick R. Veres[1,**], Jeff Peischl[1,2], Steven S. Brown[1,11], Allen H. Goldstein[5,8], Ronald C. Cohen[6,12], and Brian C. McDonald[1]

[1]NOAA Chemical Sciences Laboratory, Boulder, CO
[2]Cooperative Institute for Research in Environmental Sciences - University of Colorado Boulder
[*]Now at Department of Earth, Atmospheric and Planetary Sciences, Massachusetts Institute of Technology, Cambridge, MA, United States
[3]ORISE at Office of Research and Development, U.S. Environmental Protection Agency, Research Triangle Park, North Carolina 27711, United States
[4]NOAA Global Systems Laboratory, Boulder, CO
[5]Department of Environmental Science and Policy Management, University of California, Berkeley, Berkeley, CA 94720, United States
[6]Department of Chemistry, University of California, Berkeley, Berkeley, CA 94720, United States
[7]Department of Environmental Science and Engineering, California Institute of Technology, Pasadena, CA 91125, United States
[8]Department of Civil and Environmental Engineering, University of California, Berkeley, Berkeley, CA 94720, United States
[9]Department of Meteorology, Naval Postgraduate School, Monterey, CA 93943, United States
[10]Department of Environmental Science and Engineering, California Institute of Technology, Pasadena, CA 91125, United States
[11]Department of Chemistry, Univesity of Colorado, Boulder, Boulder, CO 80309, United States
[12]Department of Earth and Planetary Sciences, University of California, Berkeley, Berkeley, CA 94720, United States
[13]NOAA Air Resources Laboratory, College Park, MD 20740, USA
[14]Earth Resources Technology (ERT), Inc., Laurel, MD 20707, USA
[**]Now at Research Aviation Facility, National Center for Atmospheric Research
[***]Now at Department of Energy, Environmental and Chemical Engineering, Washington University in St. Louis, MI 63130, United States

We thank all reviewers for their careful reading and comments. The suggestions offered and questions raised are well taken, and we have done our best to incorporate them into the paper.

Below we respond to both common comments and individual comments. The reviewer's comments will be shown in red, our response in blue, and changes made to the paper are shown in black block quotes. Unless otherwise indicated, page and line numbers correspond to the original paper. Sections, figures, tables, or equations referenced as "R$n$" are numbered within this response; Figures, tables, and equations numbered normally refer to the numbers in the original discussion paper.

**R1 General changes**

We note that the description of the model configuration is inaccurate in two parts and we change them accordingly.

First, the boundary condition (BC) of our 4km model simulation is not constrained by the preceding 12km model simulation, instead, it is from the static BC described in McDonald et al. (2018b). The model domain is shown in Figure R1. We change Section 3 as follows:

"WRF-Chem is set up with a horizontal spatial resolution of 4km × 4km and 50 vertical layers in California **and Nevada (Figure R1)** during the summer of 2021. It is a nested domain simulation with the initial  conditions constrained by a preceding 12km × 12km model simulation covering the Contiguous US. **We use the static boundary condition described in McDonald et al. (2018b).**"

In line 448:

"In addition, in our model simulation, the LA basin experiences the background $O_3$ of 30 ppb transport outside of the model domain defined by the **static** boundary condition . "

Second, in our 4km model simulation, the boundary layer clouds are not incorporated in the photolysis calculation. We change the Section 2.2:

"To better represent the photolysis  **rate**, we incorporate  total column ozone from **the Global Forecast System (GFS) Model into the TUV scheme, consistent with** Rapid Refresh coupled with Chemistry (RAP-Chem) (Benjamin et al., 2016)."

We also note that there was a minor mis-calculation of MDA8 $O_3$. We calculated the MDA8 $O_3$ using the average of the maximum 8 hours of $O_3$ in each day, yet the correct definition of MDA8 $O_3$ is the maximum of the rolling 8-hour average of $O_3$. Therefore, we recalculate the MDA8 $O_3$ and update the corresponding Figure 3, Figure 4, and statistics in the paper. Overall, the updated MDA8 $O_3$ is lower but the impact on the comparison of MDA8 $O_3$ between observation and model simulation is marginal. For instance, in Line 282-287:

"Generally, the East basin exhibited significantly higher pollution levels compared to the West/Central LA, characterized by an average difference of  25 ppb in MDA8 $O_3$ and 2.7 $\mu$g/m$^3$ in daily $PM_{2.5}$ based on the AQS network observations. Both WRF-Chem simulations successfully reproduce the positive gradient of pollution levels between west and east LA despite an overprediction in MDA8 $O_3$ and an underprediction in daily $PM_{2.5}$. WRF-Chem with the RACM2B-VCP mechanism shows slightly better results than the RACM-ESRL-VCP mechanism with respect to NMB. The NMB in MDA8 $O_3$ is 10.7% and 9.3% and instead of, the NMB in daily $PM_{2.5}$ is -13% and -11.9% for WRF-Chem with RACM-ESRL-VCP and RACM2B-VCP mechanisms, respectively. "

[Figure]

**Figure R1.** The model domain defined in our WRF-Chem simulation at the spatial resolution of 4km, and the region in red denotes the LA domain used in our analysis.

Last, The PAN SUNVEx observations were updated since submitting this paper on the data archive: https://csl.noaa.gov/projects/sunvex/. We updated all analyses and plots as described further below to use the latest version (R1) of the PAN observations. As shown in FigureR2, the difference in PAN between observation and model simulation is much smaller. We
40   updated Figure 8 and the first paragraph of Sect. 7:

"In Section 6, we conducted a model evaluation of trace gases observed and reported in three sets of observations. Additional measurements of peroxyacetyl nitrate (PAN) and aerosols are available in the ground site measurements. While PAN is treated as a surrogate species in the RACM-ESRL-VCP mechanism, the RACM2B-VCP mechanism separates it out from the higher carbon acyl nitrates such as peroxypropionyl nitrate (PPN), enabling a direct comparison to the observations. As
45   shown in Figure 8(a), the model with the RACM2B-VCP mechanism underestimates PAN levels at Pasadena. The median PAN concentration is  **0.79** in observations and 0.48 ppb in the model, resulting in an NMDB of  **-0.37**."

[Figure]

**Figure R2.** The comparison of PAN between ground observations at Pasadena (previous observations in blue, updated observations (R1) in orange) and WRF-Chem simulation with RACM2B-VCP mechanism (green).

**R2 Review response to Reviewer 1**

The authors develop a new chemical mechanism, RACM2B-VCP for WRF-Chem that can better represent VCP sources in the urban environment. They evaluate their model against observations from the RECAP-CA airborne campaign and the SUNVEx ground and mobile lab campaign. The authors find that 52% of the VOC reactivity and 35% of the local enhancement of MDA8 ozone come from anthropogenic VOC emissions, and 50% of this is attributable to VCP emissions. This manuscript is a helpful description of model implementation and analysis of VCP emissions and chemistry and should be accepted for publication after minor revisions described below.

We appreciate the positive feedback from the reviewer and the comments are addressed below.

The authors discuss the difficulty in comparing absolute concentrations from WRF-Chem to ground site measurements. Could the authors use emission ratios or other chemical coordinates to better take advantage of these measurements and use them to constrain the performance of the VCP mechanism?

The reviewer raised a very good point that a better way to constrain the VCP mechanism is using emission ratios, however, we respectfully argue that it is out of the scope of this work. The main goal of this study is to develop a new chemical mechanism RACM2B-VCP for WRF-Chem to better represent the VOC chemistry and then validate the model simulations by comparing them against observations. A major strength of this study, compared to other modeling studies, is that we are not relying on one

observation dataset, instead, we use three independent observations, including airborne, mobile, and ground measurements. Even though comparing the absolute concentration from WRF-Chem to ground site measurements is not optimal as the ground observations are likely affected by local sources, we show the model simulations are falling within the range of these three observations.

The authors describe adding 8 VOCs to the model. The paper would be more beneficial to policy-makers and other modelers if they could discuss which VCPs are most contributing to ozone production. Is ethanol the key player? Or are there significant contributions from other species?

The VOC reactivity is a good proxy to access the contribution of each VOC species to ozone production. In Figure 6 we show the median VOC reactivity ($VOC_r$) from each calibrated species between observations and WRF-Chem simulation. From both observations and model simulation, we show that VOC species, including isoprene and its oxidation products, ethanol, and acetaldehyde, are important contributors to $VOC_r$.

Besides, from Figure 6, we show that the model underestimates the $VOC_r$ compared to the observations, mainly due to underrepresented ethanol and monoterpene. Therefore, we conclude that improving the model representation of ethanol and monoterpene is the key to yielding a better model simulation of $VOC_r$ and $O_3$ chemistry. In our study, we also show that VOC emitted from the VCP sources contributes to 5-10% of MDA8 $O_3$, which is relatively small compared to the overall $O_3$ budget. Future work will address the current biases in $VOC_r$ and use this new mechanism for more policy-relevant sensitivity tests.

It is also worth noting that running sensitivity tests in a 3D model is computationally expensive, but past work has used box-models with a similar anthropogenic emission inventory used in this work to investigate the source apportionment of individual VCP VOCs to ozone formation. For instance, Figure 4 in Coggon et al. (2021) shows that among all VCP emissions, monoterpenes, ethanol, and oxygenates, constitute ~50% of the VCP-produced ozone.

Specific Comments

Line 35 – The statement that VCPs are an 'emerging source' could be read to mean that VCPs are a new source. Is that what you mean? It is my understanding that VCPs have always existed? Or do you mean 'recently recognized' or something similar?

The "emerging source" here means that they are recently recognized as an important VOC source. To avoid the confusion, we delete the sentence in Line 35:
"Additionally,  VCP sources, including solvents, adhesives, cleaning agents, pesticides, and personal care products, constitute half of the fossil fuel VOC emissions in industrialized cities (McDonald et al., 2018b)."

Line 50 – Can you explain why this is: "reduced NOx results in higher ozone at low temperatures"?

The temperature dependence of ozone is discussed in detail in Nussbaumer and Cohen (2020). They found that when $NO_x$ is reduced, less $O_x$ is present as $NO_2$, leading to higher ozone at low temperatures. We acknowledge that this statement is confounding without further explanation, so we delete it in the context:

95    "For instance, Nussbaumer and Cohen (2020) found a lower temperature dependence of ozone in 2014-2019 than in 1997-1998 . "

Supplement – The chemistry for eucalyptol is not listed in the supplement.

We described the chemistry for eucalyptol in the main context in Line 173:

"We account for the reaction of eucalyptol with OH; the products are the same as HC8 surrogate species and the rate constant

100    is $1.1 \times 10^{-11}\ cm^3 molec^{-1} s^{-1}$ (Corchnoy and Atkinson, 1990). We also add it into Table S2 (Table R1):"

Line 189 – What is the vertical resolution of the model?

The model has 50 hybrid vertical layers up to 50 hPa. We add it in Line 189:

"WRF-Chem is set up with a horizontal spatial resolution of 4km $\times$ 4km **and 50 vertical layers** in California and Nevada during the summer of 2021. "

105    Line 209 – Are the emissions monthly? Do you apply diurnal or weekend/weekday scale factors to the emissions?

The emissions have near real-time scaling factors to address the differences in the diurnal cycle, the weekend/weekend, and months. We describe it in Line 195:

"The anthropogenic emissions here are from the FIVE-VCP-NEI17NRT inventory. This inventory is described by McDonald et al. (2018a, b) and Coggon et al. (2021) and updated by  **He et al. (2024)** with **near real-time (NRT)**

110    **scaling factors** capturing changes in emissions in the 2019-2021 timeframe, such as those due to the COVID-19 pandemic. "

Line 247 – Can you explain what you mean by micro-scale and middle-scale environments? Are these sites near am emissions source? Or do you mean to imply something about the geography?

The measurement scale is the geographic scope of the air quality measurements made by the monitor and generally provides information about how representative a site is to the broader geographical region. EPA reports the measurement scale of

115    each AQS site, including micro-scale (0-100m), middle-scale (100-500m), neighborhood scale (500m-4km) and urban scale (4-50km). It can be found by navigating to each site through the EPA AirData Viewer (https://epa.maps.arcgis.com/apps/webappviewer/index.html). We add in Line 247 for clarification:

"**Based on the measurement scale of the AQS sites reported by EPA,** one site (Ontario-Route 60 Near Road, -117.62°, 34.03°) representing the micro-scale environment and another site (Pomona, -117.75°, 34.07°) representing the middle-scale

120    environment are filtered out as WRF-Chem at 4km cannot represent the dynamics and chemistry at the spatial scale less than 2km. "

| Reaction | Reaction coefficient | Reference |
|---|---|---|
| PROG+HO=0.387 ALD+0.613 HAC+HO2 | 2.15D-11 | Coggon et al. (2021) |
| GLYCR+HO=0.45 ALD+0.55 HKET+HO2 | 5.45D-11 | |
| IPOH+HO=0.861 ACT+0.139 ACTP+0.861 HO2 | ARR2( 2.1D-12, -270.0_dp, TEMP) | |
| D4SILX+HO=ROH | 1.3D-12 | Alton and Browne (2020) |
| D5SILX+HO=ROH | 2.1D-12 | |
| PCBTF+HO=0.060 TLP1+0.763 TR2+0.177 CSL+0.177 HO2 | 2.5D-13 | Atkinson and Arey (2003) |
| PDCBZ+HO=0.352 BENP+0.118 EPX+0.530 PHEN+0.648 HO2 | 3.2D-13 | |
| **ECLP+HO=0.951 HC8P+0.025 ALD+0.024 HKET+0.049 HO2+H2O** | **1.11D-11** | Corchnoy and Atkinson (1990) |

**Table R1.** The newly added reactions addressing the VOCs from VCP sources and eucalyptol to RACM2B-VCP chemical mechanism as well as their reaction coefficients.

[Figure]

**Figure R3.** The comparison of time series of hourly O$_3$ between AQS site and two WRF-Chem simulations, one with RACM-ESRL-VCP and another one with RACM2B-VCP chemical mechanism at AQS site at Pasadena. The corresponding NMB and R$^2$ between model simulations and observations of hourly O$_3$ are shown on each plot.

Figure 2 – Can you clarify which site is in Figure 2?

The site in Figure 2 is in Pasadena, we add the longitude and latitude of the site in the caption:

"The comparison of time series of hourly O$_3$ between AQS  and two WRF-Chem simulations, one with RACM-ESRL-
125  VCP and another one with RACM2B-VCP chemical mechanism at the AQS site **located** at Pasadena (**-118.13°, 34.13°**)."

Figure 2 – It looks like WRF-Chem RACM-ESRL-VCP has higher nighttime ozone than WRF-Chem RACM2BVCP. If so, could you explain why this might be? Do you see a reduction in model nitrate aerosol formation that improves comparisons with observations?

We note that in Figure 2, we only include the observations and model simulations at 5-22 local time. Therefore, we update
130  Figure 2 (Figure R3) to include the comparison for the full day:

We also updated Table R2 to the changes in the comparison of hourly O$_3$ in both WRF-Chem simulations with RACM-ESRL-VCP and RACM2B-VCP mechanism against the AQS site measurements.

| Site name | Longitude | Latitude | RACM-ESRL-VCP | RACM2B-VCP |
|---|---|---|---|---|
| West Los Angeles | -118.46 | 34.05 | 0.05 (0.54) | 0.0 (0.52) |
| North Main Street | -118.23 | 34.07 | 0.07 (0.77) | 0.03 (0.77) |
| Compton | -118.2 | 33.90 | -0.05 (0.65) | -0.09 (0.65) |
| Signal Hill (LBSH) | -118.17 | 33.79 | -0.13 (0.54) | -0.17 (0.54) |
| Pasadena | -118.13 | 34.13 | 0.11 (0.82) | 0.08 (0.83) |
| Pico Rivera | -118.07 | 34.01 | 0.19 (0.78) | 0.15 (0.79) |
| La Habra | -117.95 | 33.92 | 0.27 (0.77) | 0.22 (0.78) |
| Anaheim | -117.94 | 33.83 | 0.20 (0.73) | 0.15 (0.74) |
| Upland | -117.63 | 34.10 | 0.21 (0.88) | 0.19 (0.90) |
| Mira Loma (Van Buren) | -117.49 | 34.00 | 0.11 (0.89) | 0.09 (0.90) |
| Rubidoux | -117.42 | 34.00 | 0.12 (0.87) | 0.09 (0.89) |
| San Bernardino | -117.27 | 34.11 | 0.12 (0.87) | 0.10 (0.88) |

**Table R2.** The comparison of hourly $O_3$ in both WRF-Chem simulations with RACM-ESRL-VCP and RACM2B-VCP mechanism against the AQS site measurements over the study time period, including June and August 2021. For each site, we calculate the normalized mean bias (NMB) and the coefficient of determination ($R^2$). The $R^2$ is shown in the parenthesis.

We then compare the diurnal cycle of $O_3$ at Pasadena from observations and model simulations, shown in Figure R4. The difference in nighttime ozone from two WRF-Chem simulations, configured with RACM-ESRL-VCP and RACM2B-VCP, respectively, is relatively small. We would need to do more sensitivity tests to fully understand this difference at night, which is challenging in a 3D model due to the computational cost of simulations, this could be caused by a lot of possible differences in the chemistry and is best explored in a box model. We will hopefully explore this further in future work.

Line 373 – The bias of 59% for D5-siloxane is high to be called 'good agreement'. Can you comment more on the possible reasons for this bias and what it might imply about the VCP inventory?

Good point. It is challenging to make comparisons between observation and simulation from the chemical transport model. As the model is at 4km, the comparison between observation and model is subject to bias due to spatial resolution and the influence of the local sources. While the normalized median bias is 59%, from Figure 5(c), the model simulated D5-siloxane is not completely out of the range of observed D5-siloxane, except for the ground measurement. Besides, the observation also embeds a large uncertainty of around 30%. Consistent with past work (McDonald et al., 2018a) we assume comparisons with individual VOCs within a factor of 2 as reasonable. Now that we have several VCP tracers in our mechanism (this work), we can use this mechanism in future work to explore how well our model represents these VCP tracers across many cities (e.g., using data from the AEROMMA campaign, which will be fully public in Sept 2024). Future work that combines this new mechanism and the AEROMMA data will have the right tools available to further refine the FIVE-VCP inventory.

[Figure]

**Figure R4.** The comparison of the diurnal cycle of $O_3$ between AQS site and two WRF-Chem simulations, one with RACM-ESRL-VCP and another one with RACM2B-VCP chemical mechanism at AQS site at Pasadena.

However, we agree with the reviewer that the statement of 'good agreement' is not clear enough, so we change it as follows: "Shown in Figure 5 (c) and (d), we find that the WRF-Chem simulation with RACM2B-VCP mechanism agrees well with airborne measurements for  PCBTF **despite an overprediction for D5-siloxane**, featuring the NMDBs of  **-5.4% for PCBTF and 59.4% for D5-siloxane**. "

Line 377 – Toluene and C8 aromatics can have a source from solvents (https://doi.org/10.5194/acp-21-6005-2021). Can you look at a weekend/weekday analysis for species you attribute to traffic or does changes in their lifetime make this too difficult?

We analyzed the weekend/weekday difference in both toluene and xylene from RECAP airborne measurements and SUN-VEX ground measurements. The lifetime of toluene is $\sim$ 2 days and the lifetime of C8 aromatic is 1 day or less (Atkinson et al., 2006), which allows us to attribute the weekend/weekday difference to emission pattern. In Figure R5, we show that both

[Figure]

**Figure R5.** The distribution of observed toluene (a,b) and xylene (c,d) during weekdays and weekends from RECAP airborne and SUNVEX ground measurements, respectively. The variation in each bin is shown by a whisker plot; the line denotes the median value.

toluene and xylene are significantly lower during weekends than on weekdays, indicating that they are predominantly emitted by traffic emissions with the same weekday-weekend pattern.

Line 378 – I have heard some discussion of anthropogenic sources of isoprene. Have you looked at whether isoprene correlates with any anthropogenic tracers to support the statement that it is solely biogenic?

This is an interesting point. Gasoline was found to be the dominant source of isoprene at night (Wernis et al., 2022), however, daytime isoprene is dominantly biogenic. Shown in Table R3, we calculate the correlation coefficients ($R^2$) between isoprene and other VOC species from three different observations used in our study. Benzene, toluene, benzaldehyde, and xylene are tracers of fossil fuel sources; D5-siloxane and PCBTF are tracers of VCP sources. Overall, we do not see a consistently strong correlation between isoprene and these anthropogenic tracers across these three observations, indicating that observed isoprene is more likely solely biogenic.

Figure 6 – It looks like the model overpredicts acetaldehyde even though it underpredicts ethanol, a major precursor, by a large amount. Is there a large primary source of acetaldehyde in the model?

|              | RECAP airborne | SUNVEX mobile | SUNVEX ground |
|--------------|----------------|---------------|---------------|
| MACR+MVK     | 0.84           | 0.42          | 0.64          |
| Benzene      | 0.30           | 0.08          | 0.24          |
| Toluene      | 0.41           | 0.07          | 0.29          |
| Benzaldehyde | 0.14           | 0.02          | 0.19          |
| Xylene       | 0.24           | 0.07          | 0.20          |
| D5-siloxane  | 0.58           | 0.00          | 0.23          |
| PCBTF        | 0.56           | 0.00          | 0.32          |

**Table R3.** The correlation coefficients ($R^2$) between observed isoprene and other VOC species from RECAP airborne, SUNVEX mobile, and ground measurements.

While ethanol is consistently underpredicted, the model only overpredicts acetaldehyde compared to the RECAP airborne measurements and underpredicts acetaldehyde compared to SUNVEX mobile and ground measurements (Figure 6 and Table 1). In the model, There are primary mobile sources of acetaldehyde while it can also be formed as a result of the photochemical oxidation of hydrocarbons. The relative contribution of primary and secondary sources of acetaldehyde needs more investi-

175 gation. However, we hypothesize that biases in OH could impact these results where high OH could lead to lower ethanol and higher acetaldehyde in the model and this would impact the comparison more for aircraft observations where air is more oxidized aloft than at the surface.

Line 414 – Should some VCPs that evaporate have a temperature dependence (such as solvents) or is this effect small? Could you discuss expectations for temperature dependence of VCPs a little more here?

180 Sure. VCP emissions may have temperature dependence due to evaporation, however, the VCP emissions are temperature-neutral in our simulation. Therefore, if the temperature dependence of VCP emission is significant, we should expect that the temperature dependence of VCP tracers is significant from observations, yet is not captured by the model simulation. As shown in Figure S7, we find that this is the case only for PCBTF, not for D5-siloxane, which is consistent with the fact that D5-siloxane is emitted from personal care products and PCBTF is emitted from solvent-based coatings. We add more discussion in Line

185 414:

" However, it is worth noting that temperature dependence is not solely attributed to emissions. For instance,  benzene are emitted from sources that are temperature-neutral in our model simulation, yet a positive temperature dependence exists in Pasadena, driven by meteorological factors. **Besides isoprene, we**

190 **find a strong temperature dependence of PCBTF from the observations but is not fully captured in the model simulations (Figure S8(e)). As PCBTF is emitted from solvent-based coatings, it may imply that the temperature dependence of some VCP sources exists due to evaporation and it is not yet included in our model.**"

[Figure]

**Figure R6.** The distribution of the ratio of PAN and acetaldehyde from ground measurement and model simulation with RACM2B-VCP chemical mechanism. The variation in each bin is shown by a whisker plot; the line denotes the median value.

Line 424 – Here could you look at PAN vs. acetaldehyde to look at whether the chemistry is producing PAN at the correct rate even if it is underestimated due to model resolution etc?

195    This is a very good point, even though we only obtained PAN observation from ground measurement at Pasadena. Compared to the ground measurement, the model simulation underpredicts PAN and acetaldehyde. With the updated ground PAN obser­vations, as shown in Figure R6, the model slightly underpredicts the PAN to acetaldehyde ratio compared against observations. The median of the PAN to acetaldehyde ratio is 0.25 from observation and 0.21 from the WRF-Chem. PAN production is highly sensitive to meteorology, for instance, temperature and downwind transport. The model resolution could be a possible reason

200    for slower PAN production, however, future work is needed to identify the cause of the discrepancy in PAN to acetaldehyde ratio between observation and model simulation.

Line 429 – Can you provide any reason for the model underestimate of nitrate aerosol, particularly given that the model correctly simulates sulfate and ammonium? Could the model NOx be too low? Or the model RH be incorrect etc? Why does the model both underestimate nitrate but also simulate negligible nitrate variability?

205    We add in the text of Sect. 7 a description of the nitrate aerosol bias when using the MADE aerosol scheme. Given the low concentrations of nitrate aerosol in the observations in Los Angeles, this model bias does not have a large impact on the total

aerosol concentrations and thus has little impact on our main conclusions in this paper. We still think it is important to evaluate all species of aerosols as done in Figure 8 in this work, so that future studies understand, for which conditions this model setup is most applicable. To further emphasize this point, we add to the main text that future work should address this nitrate aerosol bias before applying this WRF-Chem configuration in locations or seasons where nitrate aerosols comprise a larger fraction of the total aerosol budget than during the summer in Los Angeles. For further context on this issue, originally, our WRF-Chem configuration, chemistry option 108 with RACM-ESRL-VCP chemistry, was using ISORROPIA 2 as described in Li et al. (2016). However, we realized that this configuration was only configured properly for chemistry option 109 due to the formation of unrealistically large nitrate aerosols in Los Angeles in our initial simulations. Due to our analysis, in WRF-Chem version 4.4 and in this work, ISORROPIA 2 is turned off by default and the older MADE aerosol scheme is used instead for chemistry option 108 as described in the WRF-Chem user guide here(https://ruc.noaa.gov/wrf/wrf-chem/Users_guide.pdf). We will work to further improve the inorganic aerosol representation in this WRF-Chem setup in future work.

"**However, the model with the MADE aerosol scheme fails to simulate nitrate aerosols. It contradicts the study in South Korea where the MADE aerosol scheme in WRF-Chem overpredicts nitrate aerosol (Lee et al., 2020). It suggests that the MADE aerosol scheme bias for nitrate aerosol is specific to the chemical and physical conditions in a given location and season. Accurately representing inorganic nitrates is challenging for models as further described in the review by Xie et al. (2022) and references therein. Future work will address this nitrate aerosol bias further, especially when applying the WRF-Chem model over regions or seasons where nitrate aerosols comprise a larger fraction of the total aerosol budget than during the summer in Los Angeles (Figure 8).**"

Line 482 – Can you discuss the implications of the model underestimate in VOC reactivity (Fig. 6 & 7) but the overestimate in D5-siloxane on your findings in this section? What further constraints are needed to improve the inventory or the model? For example, do you think the large monoterpene underestimate is biogenic or from VCPs? Could this be a good place to discuss D5-siloxane in ratios to other species emitted from adhesives and personal care products? Or are VCP species like ethanol spread across too many sources to perform this type of analysis? Line 499 – What do you think the cause of these model underpredictions of monoterpenes, ethanol, and acetaldehyde could be?

We think the underprediction of VOC reactivity, notably ethanol and acetaldehyde, are attributed to cooking emissions. Coggon et al. (2024) shows that based on mobile laboratory observations, cooking may account for as much as 20% of the total anthropogenic VOC emissions observed by PTR-ToF-MS in Las Vegas. In contrast, emissions estimated from county-level inventories report that cooking accounts for less than 1% of urban VOCs.

The cause of underprediction in monoterpene is complicated as monoterpene is emitted from a mixture of biogenic and VCP sources. Pfannerstill et al (in review) calculated the monoterpene emissions in LA using airborne flux measurements. They found a tight correlation between monoterpenes and biogenic isoprene fluxes, while they did not observe a correlation between monoterpene emissions and any of the identified anthropogenic tracers. Therefore, we think the underprediction of monoterpene is attributed to underestimated biogenic sources.

240      We hope that the AEROMMA dataset (data made public in Sept 2024) will help to better understand these biases. And once we fully understand where the bias is coming from we can improve the model emissions, chemistry, or meteorology accordingly.

     Line 508 – The statement "Updating the biogenic emission inventory, particularly over urban regions." in the conclusions is unclear. It would be better if the authors wrote out the reasoning behind each bullet point. For example, "Our study shows we
245   need to update the biogenic emission inventory because of model underestimates in isoprene and monoterpenes"?

     Thanks for the suggestion. We updated each bullet point by adding the reasoning:
"Looking ahead, our study suggests a roadmap for further refining VOC chemistry simulations:

1. Updating the biogenic emission inventory, particularly over urban regions**, as the model significantly underestimates monoterpene**.

250   2. Implementing a more accurate SOA scheme to better characterize the source apportionment of $PM_{2.5}$**, as the model fails to reproduce the hourly variation and the temperature dependence of $PM_{2.5}$ from the AQS observation**.

3. Addressing the existing gaps between observational VOC reactivity and model predictions, thereby improving the overall model representation**, as the current model still underpredicts the VOC reactivity.**

"

255   Figure S9 – How do VOCs contribute to NOx?

     We intend to show that the changing VOC emissions alter the OH concentration in the model simulation, therefore affecting the simulated $NO_x$. However, we agree with the reviewer that "contribute" is not the right word here, and the way we define the contribution of each VOC emission to VOC reactivity and $O_3$ doesn't apply to $NO_x$. Therefore, we decide to take $NO_x$ out in Figure S10 (also Figure R7 here).

260

[Figure]

**Figure R7.** The contribution of VOC emissions from VCP and fossil fuel sources to VOC reactivity (a) and MDA8 $O_3$ (b), in West/Central LA and in East basin, under three temperature bins.

**R3 Review response to Reviewer 2**

The paper developed a RACM2B-VCP mechanism based on the RACM2Berkeley2.0 mechanism to better represent the chemistry of VOC. They evaluate the performance of RACM2B-VCP for ozone and PM2.5 by comparing WRF-Chem simulations with AQS surface network data and the previous RACM-ESRL-VCP mechanism. The RACM2B-VCP's accuracy in representing NOx, CO, VOCs, PAN, and aerosols was also investigated. The temperature dependence of ozone, the effects of VCP, biogenic and fossil fuel emissions on VOC reactivity and ozone were analyzed.

The new chemical mechanism proposed in this manuscript is meaningful for improving the simulation ability of air quality models for VOCs. However, the evidence presented in the paper to prove the superiority of the RACM2B-VCP chemical mechanism is far from convincing. The explanation for the differences in simulation results is almost missing throughout the whole manuscript. The manuscript needs to be carefully revised before it may be considered for publication.

We appreciate the reviewer's comment. However, we respectfully disagree with the reviewer that our results are not convincing. The new RACM2B-VCP chemical mechanism developed throughout our study is superior to the existing RACM-ESRL-VCP chemical mechanism not in the sense that RACM2B-VCP yields a better agreement on simulated $O_3$ compared to observations. In contrast, we argue that RACM2B-VCP chemical mechanism yields a comparable performance in simulating $O_3$ compared with RACM-ESRL-VCP chemical mechanism, as shown in Sect. 5.

More importantly, compared to the RACM-ESRL-VCP mechanism, the RACM2B-VCP mechanism offers a more complex representation of chemistry, especially on VOC chemistry, making it more suitable for assessing the chemistry accuracy, and evaluating whether the model accurately simulates ozone formation for the right reason. The RACM2B-VCP chemical mechanism has more than twice the number of species and reactions compared to the RACM-ESRL-VCP. The VOC chemistry is represented by lumped species in RACM-ESRL-VCP, making it extremely challenging to compare against the observations. However, with more independent VOC species and chemistry included in RACM2B-VCP, we can directly compare the model-simulated VOC species against observations to evaluate the model performance, as shown in Sect. 6-7. For instance, we can directly compare the following species from model simulations using RACM2B-VCP against the observation, including PAN, acetaldehyde, toluene, benzene, xylene, MACR+MVK and VCP tracers. However, these comparisons are not possible for RACM-ESRL-VCP as they are either treated as lumped species or are not included in the RACM-ESRL-VCP mechanism. As shown in Figure R8, benzene is not included as an individual species in RACM-ESRL-VCP so it cannot be compared to observations as it is in RACM2B-VCP mechanism. Besides, in RACM-ESRL-VCP, the toluene is lumped with benzene, acetaldehyde is lumped with other aldehydes, and PAN is lumped with other acetyl nitrates. If we compare the simulation of these lumped species using RACM-ESRL-VCP against observations, we would draw the wrong conclusion that the model has a much higher overprediction of toluene and has a better agreement on acetaldehyde and PAN.

The scientific significance of developing this more complex chemical mechanism RACM2B-VCP is emphasized throughout the text. In the Introduction Section, we illustrate it in Line 73-78:

"...By adding these VCP VOC tracers explicitly into the RACM2B-VCP mechanism and comparing them directly to observations, we are able to better constrain emission inventories and identify gaps in our understanding of VCP emissions and chemistry than are possible with more condensed mechanisms. Additionally, because the RACM2B-VCP mechanism is more complex than the RACM-ESRL-VCP mechanism, more tracers for mobile and biogenic emissions and their oxidation products are available to directly compare against observations, which enables a more complete evaluation of VOC emissions and chemistry in general. "

In Sect. 3 we use benzene vs toluene as an example:

"Because the RACM2B-VCP mechanism has more species, the mapping is more explicit and requires fewer scaling factors. For instance, benzene is mapped to toluene in the RACM-ESRL-VCP mechanism. We need to apply a scaling factor of 0.29 to account for the difference in OH reactivity between benzene and toluene. In contrast, benzene and toluene are treated as separate species in the RACM2B-VCP mechanism. There is no need to apply the scaling as it is in RACM-ESRL-VCP, which improves the representation of aromatic oxidation and enables a more fair evaluation against observations."

We then re-emphasize it at the beginning of Sect. 6 and Sect. 7:

"Besides a direct comparison of $O_3$, it is important to verify whether the model accurately simulates ozone formation for the right reason by evaluating modeled $O_3$ precursors against the observations. Compared to the RACM-ESRL-VCP mechanism, the RACM2B-VCP mechanism offers a more complex representation of chemistry, making it more suitable for directly comparing to observations in order to assess the chemistry accuracy."

[Figure]

**Figure R8.** The comparison of toluene, benzene, acetaldehyde, and PAN between ground observations at Pasadena (gray) and WRF-Chem simulations with RACM-ESRL-VCP (orange) and RACM2B-VCP mechanism (blue). The red shades represent the lumped species from the WRF-Chem simulation with the RACM-ESRL-VCP mechanism. The distribution is shown by a whisker plot; the black dot denotes the mean value and the line denotes the median value.

310 "... While PAN is treated as a surrogate species in the RACM-ESRL-VCP mechanism, the RACM2B-VCP mechanism separates it out from the higher carbon acyl nitrates such as peroxypropionyl nitrate (PPN), enabling a direct comparison to the observations... "

Line 212: Please explain how to add isoprene emission in the RACM-ESRL-VCP mechanism?

Following Scott and Benjamin (2003), the biogenic emissions (isoprene and monoterpene) within the urban areas are added
315 based on the urban land cover type. Shown in Scott and Benjamin (2003), urban areas in the Los Angeles basin exhibit isoprene emissions of about 1 mg m$^{-2}$ h$^{-1}$, with isolated cells showing slightly elevated emissions ranging between 3 and 4 mg m$^{-2}$ h$^{-1}$.Urban vegetation monoterpene emissions range between 0.1 and 0.2 mg m$^{-2}$ h$^{-1}$.

We note that our description is confusing here. These biogenic emissions described in Scott and Benjamin (2003) are added to both the RACM-ESRL-VCP mechanism and RACM2B-VCP mechanism. However, for the RACM2B-VCP mechanism,
320 re-speciation and adjustment are needed. We changed the Line 212-215:

"For the biogenic emission used in **both** the RACM-ESRL-VCP and RACM2B-VCP mechanism, we add additional isoprene and monoterpene emissions following Scott and Benjamin (2003) **with re-speciation for monoterpene in the RACM2B-VCP mechanism**. Among the monoterpene emissions, we assume that 20% is limonene and 80% is alpha-pinene, which is the same as the previous work over the LA region (Kim et al., 2016). **Besides, we**  update the urban biogenic emission
325  **to account for the emission of eucalyptol as included in the RACM2B-VCP mechanism**. Eucalyptol constitutes a significant portion of total monoterpene emissions, ranging from 2% and 72% depending on the tree types (Owen and Penuelas, 2013; Van Meeningen et al., 2017; Zuo et al., 2017; Purser et al., 2021). Van Rooy et al. (2021) provided the most recent biogenic VOC observations in the Los Angeles Basin and found that eucalyptol comprises 10% of total monoterpene emissions. Therefore, we adjust the ratio of monoterpene
330 emissions in accordance with Van Rooy et al. (2021), with 37% limonene, 53% alpha-pinene, and 10% eucalyptol for the RACM2B-VCP mechanism."

Line253: What time does "noontime" refer to?

"noontime" refers to 12pm local time. We update the sentence in Line 253:

"To normalize the temperature across the different campaigns, we use the **local** noontime surface temperature "

335 Line270: What is the unit of NMB?

The NMB is defined as:

$$NMB = (\frac{\bar{M}}{\bar{O}} - 1) \tag{1}$$

Where $\bar{M}$ and $\bar{O}$ are the mean of the model simulation and the observations. Therefore NMB is unitless.

Figure 4: The MDA8 O3 concentrations in RACM2B-VCP on days with lower and median temperature are lower than those in RACM-ESRL-VCP, while on days with higher temperature, RACM2B-VCP simulated higher MDA8 O3 values. Please explain the reasons for this difference.

As RACM2B-VCP and RACM-ESRL-VCP are two completely different chemical mechanisms, it is extremely challenging to identify the reason for the difference in the temperature dependence of MDA8 $O_3$. However, the difference in simulated MDA8 $O_3$ from both model simulations is 2.2 ppb on average on days with lower temperatures and 1.9 ppb on days with higher temperatures, making it even more difficult to determine the exact reasons for the differences.

Our best hypothesis is the slight difference in the biogenic emission between RACM-ESRL-VCP and RACM2B-VCP chemical mechanism. In the RACM-ESRL-VCP mechanism, the monoterpene is represented by a lumped species. However, in RACM2B-VCP, monoterpene is represented by 20% of limonene and 80% of alpha-pinene. A more complex representation of monoterpene chemistry in RACM2B-VCP leads to the difference in the simulated temperature dependence of MDA8 $O_3$.

Line 325: The comparison of the model evaluations of NOx, CO, and VOCs by RACM-ESRL-VCP and RACM2B-VCP should be presented simultaneously.

Please refer to our response to the reviewer's major comment (R3). RACM-ESRL-VCP does not have individual trace gases such as D5-siloxane, PCBTF, toluene, or acetaldehyde, so we cannot perform the same comparison against observations as for RACM2B-VCP chemical mechanism. In terms of $NO_x$ and CO, model simulation with both chemical mechanisms show very similar results, as shown in Figure R9.

Line346: The author said, "Overall, we show that the R2 is generally higher between model simulation and airborne measurements". However, the NMDB and R2 values shown in Table 1 and Figure 5 didn't suggest that the simulation with the RACM2B-VCP mechanism is ideal. Please supplement a comparison of model simulation evaluation results with previous studies.

The statement in Line 346 compares the correlation coefficients against three observations. Compared to the R2 from model simulations to either mobile or ground measurements, the R2 between model simulation and airborne measurement is higher as airborne measurements are less influenced by local sources. The species that have much higher R2 values for the RECAP airborne campaign than the SUNVEx mobile and ground campaigns include NOx, CO, calibrated VOCr, PCBTF, CH4, methanol, ethanol, acetone, monoterpene, benzene, toluene, benzaldehyde, and xylene. Only D5-siloxane, isoprene, acetaldehyde, MVK + MACR do not follow this trend. We updated Line 346 for clarification:

"**Overall, we show that the $R^2$ is generally higher between model simulation and airborne measurements as compared to between model simulation and ground or mobile measurements.**  "

[Figure]

**Figure R9.** The comparison of $NO_x$ (a), CO (b) between observations and two WRF-Chem simulations configured with RACM-ESRL-VCP and RACM2B-VCP chemical mechanism, respectively. The comparison to the RECAP airborne measurement in yellow shade; SUNVEx mobile measurements in red shade and SUNVEx ground measurements in green shade. The distribution is shown by a whisker plot; the black dot denotes the mean value and the line denotes the median value.

Compared to existing modeling studies, a major strength of this study is that we are not relying on one observation dataset, instead, we use three independent observations, including airborne, mobile, and ground measurements. Given the uncertainty in the observations, we show that it is more robust to conduct the model evaluation by showing whether or not the model simulations fall within the range of these three observations.

In sections 5 to 7, the author only presented a comparison between observation and simulation results, without providing any explanation for the reasons for simulation bias, nor did they present any comparison with previous studies to show the superiority of the RACM2B-VCP mechanism in simulating VOC chemistry, ozone, and other species.

Please refer to our response to the reviewer's major comment (R3).

Table 1: The NMDB of acetaldehyde was different from other VOC species and significantly higher in RECAP atmosphere than that in SUNVEx mobile and SUNVEx ground. Why?

We show that the model overpredicts acetaldehyde compared to RECAP airborne measurement, yet underpredicts acetaldehyde compared to SUNVEX mobile and ground measurement. The exact reason for this discrepancy remains unknown. We have tried our best to identify the cause of the discrepancy. Our first hypothesis is the interference of ethanol in acetaldehyde observations from PTR-ToF-MS instrument (Coggon et al., 2024). We checked with the observationalists; the RECAP airborne measurements are corrected for the ethanol interference. For the SUNVEX measurements, the ethanol interference on acetaldehyde was small at Caltech (< 2%). However, there may be larger biases in the measurements due to inferences from the fragmentation of glycols, such as ethylene glycol. This interference is uncertain and there is currently no effective approach to quantify this interference. Our second hypothesis is that the discrepancy could be attributed to influences from local sources identified in the mobile and ground measurements. Future work is needed to enhance the model representation of acetaldehyde. We hope that AEROMMA observations, which will be publicly available in summer 2024 will help to resolve this discrepancy.

Figure S8: It seems that the nitrate concentration simulated by the model is particularly low. What is the reason for this poor performance?

We add in the text of Sect. 7 a description of the nitrate aerosol bias when using the MADE aerosol scheme. Given the low concentrations of nitrate aerosol in the observations in Los Angeles, this model bias does not have a large impact on the total aerosol concentrations and thus has little impact on our main conclusions in this paper. We still think it is important to evaluate all species of aerosols as done in Figure 8 in this work, so that future studies understand, for which conditions this model setup is most applicable. To further emphasize this point, we add to the main text that future work should address this nitrate aerosol bias before applying this WRF-Chem configuration in locations or seasons where nitrate aerosols comprise a larger fraction of the total aerosol budget than during the summer in Los Angeles. For further context on this issue, originally, our WRF-Chem configuration, chemistry option 108 with RACM-ESRL-VCP chemistry, was using ISORROPIA 2 as described in Li et al. (2016). However, we realized that this configuration was only configured properly for chemistry option 109 due to the formation of unrealistically large nitrate aerosols in Los Angeles in our initial simulations. Due to our analysis, in WRF-Chem version 4.4 and in this work, ISORROPIA 2 is turned off by default and the older MADE aerosol scheme is used instead for chemistry option 108 as described in the WRF-Chem user guide here(https://ruc.noaa.gov/wrf/wrf-chem/Users_guide.pdf). We will work to further improve the inorganic aerosol representation in this WRF-Chem setup in future work.

"**However, the model with the MADE aerosol scheme fails to simulate nitrate aerosols. It contradicts the study in South Korea where the MADE aerosol scheme in WRF-Chem overpredicts nitrate aerosol (Lee et al., 2020). It suggests that the MADE aerosol scheme bias for nitrate aerosol is specific to the chemical and physical conditions in a given location and season. Accurately representing inorganic nitrates is challenging for models as further described in the review by Xie et al. (2022) and references therein. Future work will address this nitrate aerosol bias further especially when applying the WRF-Chem model over regions or seasons where nitrate aerosols comprise a larger fraction of the total aerosol budget than during the summer in Los Angeles (Figure 8).**"

Figure S9: Why do biogenic VOCs contribute so much to the concentration of NOx?

We intend to show that the changing VOC emissions alter the OH concentration in the model simulation, therefore affecting the simulated $NO_x$. However, we agree with the reviewer that "contribute" is not the right word here, and the way we define

the contribution of each VOC emission to VOC reactivity and $O_3$ doesn't apply to $NO_x$. Therefore, we decide to take $NO_x$ out

415    in Figure S10 (also Figure R7 here).

Please unify the font of the figures in the main manuscript.

The font of the figures is unified by remaking the Figure 5-7.